# Probiotic Sheep Milk: Physicochemical Properties of Fermented Milk and Viability of Bacteria Under Simulated Gastrointestinal Conditions

**DOI:** 10.3390/nu17213340

**Published:** 2025-10-24

**Authors:** Małgorzata Pawlos, Katarzyna Szajnar, Agata Znamirowska-Piotrowska

**Affiliations:** Department of Dairy Technology, Faculty of Technology and Life Sciences, University of Rzeszów, Ćwiklińskiej 2D, 35-601 Rzeszów, Poland; kszajnar@ur.edu.pl (K.S.); aznamirowska@ur.edu.pl (A.Z.-P.)

**Keywords:** probiotic bacteria, in vitro digestion, whey protein isolate, soy protein isolate, pea protein isolate, functional food, physicochemical properties, microbiological analysis

## Abstract

Background/Objectives: Within the spectrum of lactic acid bacteria, *Lacticaseibacillus casei* and *Lactobacillus johnsonii* are of particular technological and nutritional significance. Protein fortification of fermented dairy systems offers dual benefits: it improves product quality while enhancing probiotic resilience. Supplementary proteins supply bioavailable nitrogen and peptides that stimulate bacterial metabolism and contribute to a viscoelastic gel matrix that buffers cells against gastric acidity and bile salts. The aim of this study was to clarify the functional potential of such formulations by assessing probiotic survival under in vitro digestion simulating oral, gastric, and intestinal phases. Methods: Sheep milk was fermented with *L. casei* 431 or *L. johnsonii* LJ in the presence of whey protein isolate (WPI), soy protein isolate (SPI), or pea protein isolate (PPI) at concentrations of 1.5% and 3.0%. Physicochemical parameters (pH, titratable acidity, color, syneresis), organoleptic properties, and microbiological counts were evaluated. The viability of *L. casei* and *L. johnsonii* was determined at each digestion stage, and probiotic survival rates were calculated. Results: Samples with *L. johnsonii* consistently exhibited lower pH values compared to *L. casei*. Across both bacterial strains, the addition of 1.5% protein isolate more effectively limited syneresis than 3.0%, regardless of protein type. Samples fortified with WPI at 1.5% (JW1.5) and 3.0% (JW3.0) were rated highest by the panel, demonstrating smooth, homogeneous textures without grittiness. The greatest bacterial survival (>70%) was observed in WPI-fortified samples (JW1.5, JW3.0) and in SPI-fortified JS3. Conclusions: Protein isolates of diverse origins are suitable for the enrichment of fermented sheep milk, with 1.5% supplementation proving optimal. Such formulations maintained desirable fermentation dynamics and, in most cases, significantly improved the survival of *L. casei* and *L. johnsonii* under simulated gastrointestinal conditions, underscoring their potential in the development of functional probiotic dairy products.

## 1. Introduction

Sheep milk represents a relatively small fraction of global dairy production, with approximately 10 million tons produced annually, yet its nutritional and technological attributes have placed it at the center of growing scientific and industrial interest [1]. In Europe, which accounts for the majority of production, sheep milk has traditionally been used for cheese manufacture, but in recent years its application has extended toward fermented milk and functional foods [1,2]. This diversification reflects broader consumer demand for non-bovine alternatives that combine nutritional density with health-promoting properties [2]. Compared with bovine milk, sheep milk contains higher concentrations of protein (5–6%), fat (6–7%), and essential minerals such as calcium and phosphorus, as well as significant levels of bioactive compounds [3]. Its compositional profile, including abundant casein and phosphate salts, imparts strong buffering capacity and confers desirable technological behavior during fermentation, yielding products with enhanced viscosity, gel firmness, and reduced syneresis [3,4]. These qualities are not only relevant for product acceptability but also create a microenvironment that supports probiotic stability during storage and gastrointestinal transit [5].

Relative to cow’s and goat’s milk, sheep’s milk exhibits higher protein and fat contents and a denser mineral profile (notably Ca and P), compositional features that underpin a greater intrinsic buffering capacity and distinct gelation behavior during fermentation [6]. Recent comparative work shows that acid gelation functionality of goat milk is generally inferior to that of cow and sheep milks, whereas sheep milk forms stronger acid and rennet gels, consistent with its higher casein and ash contents [7]. At the gastrointestinal level, inter-species differences in apparent small-intestinal amino-acid digestibility have been documented, further supporting that matrix composition and buffering co-determine proteolysis and probiotic performance [8]. In probiotic beverage models prepared from cow, sheep, and goat whey protein concentrates, matrix-dependent differences in physicochemical properties and microbial outcomes were likewise observed, underscoring the practical relevance of these compositional contrasts [9].

The integration of probiotics into dairy matrices has become a cornerstone of functional food innovation. For probiotics to exert health effects, sufficient viable cells must be delivered at the time of consumption, generally accepted as >10^6^ CFU g^−1^ [10]. Fermented sheep milk provides a promising vehicle in this respect, combining nutrient density with a matrix that favors probiotic viability. Recent studies indicate that dairy composition can significantly influence survival during gastrointestinal stress, with protein- and mineral-rich substrates offering improved protection [11]. Moreover, fortification of milk with additional proteins, such as whey protein concentrate, collagen peptides, or plant protein isolates, has been demonstrated to reinforce gel structure, reduce whey separation, and enhance probiotic resilience during simulated digestion [12,13]. These fortification strategies not only improve nutritional profiles but also provide structural advantages that may encapsulate probiotics within the gel matrix, delaying exposure to gastric acid and bile salts [13].

Within the spectrum of lactic acid bacteria, *Lacticaseibacillus casei* and *Lactobacillus johnsonii* are of particular relevance. *L. casei* has been widely investigated for its technological adaptability and health-promoting effects, including antioxidant activity, immunomodulation, and modulation of metabolic parameters [14]. Mechanistic studies demonstrate that *L. casei* can reinforce epithelial tight junction integrity by upregulating proteins such as occludin and ZO-1 (zonula occludens-1), thereby reducing intestinal permeability [15]. It also interacts with the gut-associated lymphoid tissue (GALT), stimulating anti-inflammatory cytokines such as interleukin-10 (IL-10) while downregulating pro-inflammatory mediators including tumor necrosis factor alpha (TNF-α) and interleukin-6 (IL-6), leading to attenuation of intestinal inflammation [16]. *L. johnsonii*, frequently isolated from the gastrointestinal tract of humans and animals, exhibits complementary properties. It strongly adheres to intestinal epithelial cells, facilitating colonization and competitive exclusion of pathogens [17]. Recent reports highlight its role in promoting mucin production and goblet cell regeneration, mediated by the induction of interleukin-22 and activation of host protective pathways [18]. Furthermore, supplementation with sheep-derived *L. johnsonii* strains has been shown to enhance antioxidant enzyme activity (superoxide dismutase, catalase, glutathione peroxidase), strengthen barrier integrity through increased expression of tight junction proteins, and reduce systemic inflammation in vivo [19]. In experimental colitis models, *L. johnsonii* reduced pro-inflammatory cytokines (interleukin-1 beta—IL-1β, IL-6, TNF-α, C-X-C motif chemokine ligand 1—CXCL1) while elevating IL-10, and restored the regulatory T cell (Treg)/T helper 17 (Th17) balance, underscoring its capacity to modulate both innate and adaptive immunity [20]. Genomic analyses of selected strains further reveal adaptation to gastric acid and bile salts, production of antimicrobial peptides, and the presence of stress-response genes that collectively explain its resilience in the gastrointestinal tract [21]. Together, these findings support the positioning of *L. johnsonii* as a probiotic with strong mechanistic evidence for barrier protection, immune modulation, and pathogen control.

Protein fortification provides additional opportunities to enhance both product quality and probiotic survival, i.e., viability during refrigerated storage and survival under simulated gastrointestinal conditions. Whey protein isolate (WPI) is widely used to fortify yogurts, yielding firmer gels, higher viscosity, and reduced syneresis while improving bacterial viability during storage [22,23]. Increasingly, plant proteins such as soy protein isolate (SPI) and pea protein isolate (PPI) are incorporated into dairy products, reflecting consumer demand for environmentally sustainable and nutritionally diverse protein sources [24,25,26]. Soy protein contributes essential amino acids and bioactive compounds that support metabolic health, including improved glucose regulation and type 2 diabetes management [24,26]. Pea protein, though less studied, provides a complementary amino acid profile and functional properties that improve texture, creaminess, and probiotic growth [27]. Fortification of milk with SPI and PPI has been associated with improved protein digestibility, reduction in anti-nutritional factors during fermentation, and the release of bioactive peptides [27,28]. Importantly, protein enrichment enhances probiotic performance: added proteins supply nitrogen sources and peptides that stimulate lactic acid bacterial metabolism and provide a protective gel structure that buffers cells against gastric acidity and bile salts [29].

In our previous study [30], addition of whey, soy, or pea protein isolates modulated syneresis, texture, color, and probiotic counts in fermented milk, indicating that protein enrichment can reshape gel microstructure and the microenvironment relevant to probiotic performance. Recent reports are consistent with this pattern. In cow’s-milk yogurts, fortification with high-milk-protein preparations (casein- and whey-rich) increased firmness, reduced whey separation, and supported lactic cultures compared with non-enriched controls, suggesting that protein addition can concurrently improve texture and microbial robustness [12]. Likewise, fortification with modified fermented whey protein has been shown to enhance water-holding capacity, apparent viscosity, and sensory quality—features associated with more cohesive gel networks and improved buffering around embedded bacteria [31]. Complementary work further indicates that coupling protein matrices with structured interfaces or prebiotic components can stabilize set-yogurt structures and favor probiotic performance, underscoring that protein source, aggregation state, and intrinsic buffering/mineral profiles jointly influence recovery of cells after gastric stress [32].

Although evidence has largely been generated in bovine or mixed dairy systems, the high solids and buffering capacity of sheep milk provide an advantageous basis for protein fortification. Most probiotic fortification studies have focused on bovine matrices or plant-based alternatives and typically examined a single protein source. Although ovine and caprine systems have been explored separately, to our knowledge no study has systematically integrated both animal (whey) and plant (soy/pea) protein isolates within a single fermented sheep-milk model while concurrently assessing physicochemical quality and probiotic survival across a standardized in vitro digestion. The aim of this study was to evaluate the effect of protein isolates fortification on the physicochemical and organoleptic properties of fermented sheep milk and on the survival of probiotic bacteria during simulated gastrointestinal digestion. In this study, ovine milk was enriched with whey, soy, and pea protein isolates prior to fermentation with *L. casei* 431 and *L. johnsonii* LJ, with the goal of producing high-protein, probiotic-rich fermented milks characterized by improved physicochemical stability, organoleptic acceptability, and microbial resilience. To further clarify the functional potential of these formulations, bacterial survival was examined under standardized static in vitro digestion simulating oral, gastric, and intestinal phases. By integrating nutritional and technological perspectives, this work seeks to provide new evidence for the design of sheep milk–based functional foods that couple enhanced product quality with reliable probiotic delivery.

## 2. Materials and Methods

### 2.1. Materials

Organic sheep milk I Love My Sheep (Leeb Biomilch GmbH, Wartberg an der Krems, Austria; 6% fat, 4.7% lactose, 4.9% protein, 0.16% salt) was used as the base for fermented milk production. Soy protein isolate (SPI), pea protein isolate (PPI), and whey protein isolate (WPI) were used as fortifying protein additives; all protein isolates were food-grade powders obtained from F.H.U. “KDJ” s.c. (Tarnów, Poland). Two probiotic strains were selected as starter cultures: *Lacticaseibacillus casei* 431 and *Lactobacillus johnsonii* strain LJ (Delvo^®^Pro). *L. casei* 431 was obtained from Chr. Hansen (Hørsholm, Denmark), and *L. johnsonii* LJ was sourced from DSM (Delft, The Netherlands). De Man, Rogosa and Sharpe (MRS) agar and 0.1% peptone water for microbiological analyses were purchased from Biocorp (Warsaw, Poland). All enzymes and reagents for the in vitro digestion assay were of analytical grade. The following enzymes were used: α-amylase (heat-stable, ~25,000 U/mL; Sigma-Aldrich, St. Louis, MO, USA), porcine mucin (type II; Sigma-Aldrich), pepsin from porcine gastric mucosa (≥250 U/mg; Sigma-Aldrich, St. Louis, MO, USA), bile extract (porcine; Sigma-Aldrich, St. Louis, MO, USA), and pancreatin from porcine pancreas (8× USP; Sigma-Aldrich, St. Louis, MO, USA). Buffer components and chemicals were obtained from Chempur (Piekary Śląskie, Poland): sodium dihydrogen phosphate (Na_2_HPO_4_, ≥99%), dipotassium hydrogen phosphate (K_2_HPO_4_, ≥98%), sodium chloride (NaCl, ≥99.9%), hydrochloric acid (12 M HCl), and sodium hydroxide (1 M NaOH).

### 2.2. Fermented Sheep Milk Preparation

Sheep milk was fortified with each protein isolate at two concentration levels (1.5% and 3% *w*/*w*). Separate formulations were prepared for each protein type (WPI, SPI, PPI) at each inclusion level, along with a control samples containing no added protein isolate (Table 1). Each milk mixture was thoroughly blended to disperse the added protein, then subjected to homogenization at 60 °C to ensure uniform consistency. Next, the milk formulations were heat-treated at 85 °C for 10 min.

The probiotic starter cultures were prepared by reactivating the freeze-dried bacteria in sterile growth medium prior to fermentation. Each culture was pre-activated in sheep milk at 40 °C for 5 h [33] to achieve a high viable cell count at inoculation. Once the milk samples had cooled to 37 °C, they were inoculated with the probiotic cultures at an inclusion rate of 5% (*w*/*w*). Each formulation was inoculated with *L. casei* 431 or *L. johnsonii* LJ, to obtain fermented products containing a single probiotic strain (Table 1). The inoculum was thoroughly mixed into the milk, and the inoculated mixtures were immediately dispensed into sterile 100 mL food-grade plastic containers.

Fermentation was carried out in an incubator (Cooled Incubator ILW 115, POL-EKO Aparatura, Wodzisław Śląski, Poland) at 37 °C until the pH of the milk dropped to 4.60 ± 0.10. Upon completion of fermentation, all samples were rapidly cooled to 5 ± 1 °C and then stored at this refrigeration temperature for 7 days. All fermentations for each formulation and strain were performed in triplicate (three independent batches for each experimental variant).

### 2.3. Acidity

The pH of each sample was measured directly in a 100 mL aliquot using a FiveEasy™ pH meter equipped with an InLab^®^ Solids Pro-ISM electrode (Mettler-Toledo, Greifensee, Switzerland), after cooling following heat treatment (pH 1) and on the 7th day of storage (pH 7).

Titratable acidity (TA) was determined according to the ADPI Method #007a [34], with minor modifications. In brief, 25 g of fermented milk was titrated with 0.1 N NaOH in the presence of 0.5 mL of phenolphthalein indicator, until the first faint pink coloration persisted for 30 s (indicating the titration endpoint). The TA was expressed as percentage of lactic acid (*w*/*v*) and calculated as:$$\mathrm{TA}\%=\frac{mL\;of\;NaOH\;titrant\times NaOH\;normality\times 9.008}{sample\;weight,g}$$

### 2.4. Syneresis

Syneresis (whey separation) was evaluated by measuring the amount of whey expelled from the sample after centrifugation [33]. A 10 g of fermented milk was placed in a 50 mL polypropylene tube and centrifuged at 3160× *g* for 10 min at 5 °C in a refrigerated centrifuge (LMC-4200R; Biosan SIA, Rīga, Latvia). The separated whey was collected and weighed, and syneresis was calculated as the percentage of the original sample mass represented by the supernatant.

### 2.5. Color

The color of the fermented milk was analyzed instrumentally using a portable Precision Colorimeter (Model NR 145, Shenzhen, China) based on the CIE Lab color space. Before measurements, the device was calibrated against a standard white reference tile [33].

For each sample, the colorimeter provided values for lightness and chromatic coordinates. Lightness (L*) was measured on a scale from 0 (black) to 100 (white). The a* coordinate (green–red axis) ranges from negative values (indicating green) to positive values (indicating red), and the b* coordinate (blue–yellow axis) ranges from negative (indicating blue) to positive (indicating yellow). From these primary readings, the color saturation (chroma, C*) and hue angle (h°) were calculated to describe the intensity and hue of the sample’s color, respectively. Each measurement was performed by filling a plastic cup with ~100 mL of the fermented milk and placing the colorimeter probe directly on the sample surface to record the L*, a*, b*, C*, and h° values.

### 2.6. Organoleptic Evaluation

An organoleptic evaluation was carried out in a dedicated sensory analysis laboratory using a trained panel of 30 people (15 women and 15 men, aged 22–55 years). Each panelist was seated in a separate sensory booth and provided with an evaluation form and three samples of fermented milk coded with random three-digit numbers. Panelists assessed the samples one by one in a randomized order, using both smell and taste, and recorded their impressions on a nine-point intensity scale. This scale was anchored such that 1 corresponded to the lowest intensity (or least characteristic expression of an attribute) and 9 corresponded to the highest intensity (most characteristic expression). For the 9-point hedonic scale, overall sensory acceptability was interpreted as marketable when the mean overall liking was ≥6.0. Scores ≥ 7.0 were interpreted as indicative of high acceptability [35,36].

Eight specific sensory attributes were examined for each sample [37,38]: consistency, milky-creamy taste, sour taste, sweet taste, off-taste, fermentation odor, sour odor, and off-odor. The attributes were defined for panelists according to standard descriptive terminology [39] as follows:

Milky-creamy taste: a taste reminiscent of milk powder.

Sour taste: the acidic taste associated with lactic acid.

Sweet taste: the taste associated with sucrose.

Off-taste: any atypical or uncharacteristic taste that is not normally present.

Sour odor: an odor attributable to acids.

Off-odor: any atypical or uncharacteristic odor.

### 2.7. In Vitro Digestion Simulation

After seven days of refrigerated storage, the fermented sheep milk samples were subjected to a three-step static in vitro digestion model simulating sequential conditions of the human mouth, stomach, and small intestine. The digestion procedure followed established protocols for simulated gastrointestinal transit with appropriate modifications for fermented dairy matrices [33,40,41,42].

All digestion steps were performed under strictly controlled temperature and pH conditions to ensure reproducibility. The temperature was continuously maintained at 37 ± 0.5 °C using a thermostatically controlled shaking water bath (Orbital Shaker-Incubator ES 20, Biosan, Riga, Latvia). The pH of each digestion vessel was measured and recorded at the beginning and end of every phase using a calibrated pH electrode (InLab^®^ Solids Pro-ISM electrode, Mettler-Toledo, Greifensee, Switzerland) connected to a FiveEasy™ pH meter (Mettler-Toledo, Greifensee, Switzerland).

Oral phase: An aliquot of 50 g of each fermented milk sample was transferred into a sterile 100 mL glass vessel, and 5 mL of simulated saliva was added. The simulated salivary fluid was prepared by dissolving 2.38 g Na_2_HPO_4_, 0.19 g K_2_HPO_4_, and 8.00 g NaCl in 1 L of distilled water, then adding porcine mucin (0.1 g/L) and α-amylase (150 mg/L, providing ~200 U/L of amylolytic activity). The sample–saliva mixture was adjusted to pH 6.75 ± 0.20 using 1 M NaOH to mimic the slight alkalinity of human saliva. The mixture was then incubated at 37 °C for 2 min in a shaking water bath with gentle agitation (90 rpm) to simulate oral processing.

Gastric phase: Upon completion of the oral phase, the samples were immediately subjected to gastric simulation. Porcine pepsin was added at a concentration of 13.08 mg per vessel (to achieve the enzyme activity typical of gastric juice), and the pH of the mixture was lowered to 2.0 ± 0.2 with 12 M HCl to reproduce the acidic stomach environment. The vessels were then incubated at 37 °C for 2 h in the shaking water bath (90 rpm) to allow pepsin-mediated proteolysis under continuous agitation.

Intestinal phase: After gastric digestion, the chyme was adjusted to conditions simulating the small intestine. A fresh pancreatin–bile solution was prepared, containing pancreatin (4 g/L) and bile salts (25 g/L) in distilled water. An aliquot of 5 mL of this pancreatin/bile solution was added to each digestion vessel. The pH was then raised to 7.00 ± 0.20 with 1 M NaOH, neutralizing the gastric acid and creating a near-neutral environment typical of the duodenum. The samples were incubated for a further 2 h at 37 °C with continuous mixing (90 rpm) to simulate intestinal digestion.

Each phase of the in vitro digestion was terminated at its specific endpoint, and the obtained digesta were directly subjected to microbiological assessment according to the procedure described in Section 2.8.

### 2.8. Microbiological Analysis

The stability and survival of *L. casei* 431 and *L. johnsonii* LJ were assessed both after storage and throughout the in vitro digestion. Viable cell counts of the probiotic bacteria were determined after 7 days of cold storage (prior to digestion) and after each phase of the simulated digestion (oral, gastric, and intestinal) to evaluate survival through the GI tract conditions [33].

For each analysis point, samples were appropriately diluted and plated to enumerate the viable probiotic cells. Specifically, 10 g of fermented milk or digested sample was aseptically blended with 90 mL of sterile 0.1% peptone water. The homogenate was then serially diluted (10-fold dilutions in 0.1% peptone water), and duplicate aliquots of suitable dilutions were pour-plated in MRS agar. MRS plates were incubated under anaerobic conditions at 37 °C for 72 h in a vacuum desiccator utilizing the GENbox anaer system (Biomerieux, Warsaw, Poland) and an incubator (Cooled Incubator ILW 115, POL-EKO Aparatura, Wodzisław Śląski, Poland). After incubation, colonies were counted using a colony counter (TYPE J-3, Chemland, Stargard Szczeciński, Poland), and the results were expressed as log_10_ colony-forming units per gram of sample (log CFU g^−1^).

The survival rate of the probiotics during in vitro digestion was calculated by comparing the viable counts after each digestion stage to the initial count before digestion, according to the following equation [33]:$$\mathrm{Survival}\;\mathrm{rate}\;(\%)=\frac{Viable\;counts\;in\;digested\;sample}{Viable\;counts\;in\;undigested\;sample}\times 100$$

### 2.9. Statistical Analysis

Results are reported as mean ± standard deviation (SD). Statistical analyses were performed in Statistica v13.1 (StatSoft, Tulsa, OK, USA). Physicochemical measurements were obtained in triplicate (*n* = 3 biological replicates per formulation), each measured in five technical replicates per trial; technical replicates were averaged within each biological replicate prior to analysis. Sensory data were collected from 30 assessors (*n* = 30; 15 female, 15 male) in each of three independent experimental trials (biological replicates), using a 9-point hedonic scale. Results are presented as mean ± SD of the individual assessor ratings for each formulation. Microbiological counts were generated for *n* = 3 biological replicates per formulation at each analysis, with duplicate plates per dilution; duplicate plates were averaged within each biological replicate. One-way and factorial ANOVA were used to evaluate treatment effects across all measured parameters, including individual sensory attributes. Tukey’s HSD was applied for multiple mean comparisons (*p* ≤ 0.05). Pearson correlation coefficients (r) were calculated between selected physicochemical, sensory, and microbiological variables, with statistical significance set at *p* ≤ 0.05.

## 3. Results and Discussion

### 3.1. Physicochemical Properties of Fermented Sheep Milk

To determine the impact of protein isolates on milk acidity, pH was measured after heat treatment of sheep milk with added isolates, prior to bacterial inoculation (Table 2 and Table 3). The addition of 1.5% and 3% WPI significantly decreased the pH of sheep milk (*p* ≤ 0.05), with values reduced by 0.13 to 0.90 units. In comparison, PPI lowered the pH to a lesser extent, by only 0.05–0.06 units. Notably, SPI had no statistically significant effect on milk acidity after pasteurization.

The observed reduction in pH following whey protein supplementation may be attributed to its higher content of acidic amino acids—namely, aspartic acid (10.36%) and glutamic acid (17.04%)—compared to pea protein isolate (8.77% and 12.88%, respectively) and soy protein isolate (8.42% and 13.54%, respectively). This effect is particularly relevant when considering the balance with basic amino acids such as arginine, histidine, and lysine, whose total content in the isolates is as follows: pea protein—14.00%, soy protein—14.51%, and whey protein—14.00% [43,44].

All sheep milk samples fermented with *L. johnsonii* exhibited lower pH values than their counterparts fermented with *L. casei*. This was confirmed by ANOVA analysis (Table 4), which demonstrated that the bacterial strain (*p* ≤ 0.01), the interaction between strain and isolate dose (*p* ≤ 0.01), as well as the three-way interaction between isolate type, dose, and strain (*p* ≤ 0.01), had a significant effect on the final pH of fermented milk.

The control sample CJ (milk fermented with *L. johnsonii* without protein addition) had a significantly higher pH (*p* ≤ 0.05) compared to all samples with added isolates (JW1.5, JW3, JS1.5, JS3, JP1.5, JP3). The lowest pH was observed in samples JB1.5 and JB3 (pea protein isolate fermented with *L. johnsonii*), especially in comparison to the CJ control.

In contrast, sheep milk fermented with *L. casei* and enriched with protein isolates showed higher pH values than the corresponding control CC, except for sample CB1.5. The highest pH was recorded in CS3 (milk with 3% SPI). In *L. casei*-fermented milk, increasing the concentration of soy and pea protein isolates from 1.5% to 3% significantly decreased the pH (CS1.5, CS3, CB1.5, CB3). On the other hand, in milk fermented with *L. johnsonii*, an increase in SPI to 3% significantly decreased pH (*p* ≤ 0.05), whereas an opposite trend was observed in JW3 (3% whey protein), where pH was significantly higher than in JW1.5.

In the study by Pelaes Vital et al. [45], it was observed that the time required for yogurt to reach the target pH of 4.6 was influenced by the type of added ingredients. A slightly higher growth of both microorganisms was reported in low-fat yogurts containing soy-based components, which the authors attributed to the shorter incubation period of these samples. This effect may be linked to the prebiotic properties of soy constituents, as they provide galactooligosaccharides (GOS), including non-digestible carbohydrates such as raffinose [46]. Moreover, dietary fiber has been shown to accelerate acidification during yogurt manufacture [47]. Additional studies have also confirmed that prebiotics can promote the growth of *Lactobacillus* strains [48].

Analysis of the acidity of sheep milk supplemented with protein isolates and fermented by *L. johnsonii* or *L. casei* showed that samples with *L. johnsonii* exhibited higher titratable acidity. The addition of protein isolates further increased acidity in *L. johnsonii* samples compared with the CJ control. However, only increasing the dose of PPI from 1.5% to 3% significantly enhanced acidity (by 0.51%).

In sheep milk fermented with *L. casei*, the highest acidity was observed in samples containing 1.5% WPI, although this value did not differ significantly from the CC control. Increasing the isolate concentration from 1.5% to 3% in *L. casei* samples did not result in higher acidity.

Excessively thin consistency and the occurrence of syneresis are considered major defects of yogurt [49]. To counteract these defects, stabilizers or milk-derived solids are commonly added in industrial practice to increase total dry matter. Therefore, in this study it was also investigated whether protein isolates of different origins could reduce syneresis.

Syneresis in the CJ control sample fermented with *L. johnsonii* was 18.12% higher than in the CC control fermented with *L. casei*. In both *L. casei* and *L. johnsonii* samples, a 1.5% addition of protein isolate more effectively reduced syneresis compared with 3%, regardless of isolate type. The addition of 1.5% WPI reduced syneresis by 35.83% in JW1.5 and by 6.43% in CW1.5 compared to their respective controls (CJ and CC). Similarly, 1.5% PPI decreased syneresis by 31.69% in JB1.5 and by 11.18% in CB1.5 relative to controls. Conversely, increasing WPI from 1.5% to 3% raised syneresis by 12.07% in JW3 and by 4.59% in CW3. A reduction in syneresis was also observed with 3% PPI—by 3.60% in JB3 and 6.57% in CB3—compared to JB1.5 and CB1.5, respectively.

ANOVA analysis (Table 4) confirmed that syneresis was significantly affected by isolate dose (*p* ≤ 0.01), by interactions between isolate type and bacterial strain (*p* ≤ 0.01), as well as by interactions between bacterial strain and isolate dose (*p* ≤ 0.01).

In the study by Pelaes Vital et al. [45], on day 1 the control yogurt, as well as yogurts supplemented with whey protein and soy protein, exhibited higher syneresis (*p* < 0.05) compared with yogurts fortified with milk proteins and soy flour. Furthermore, several authors have emphasized the gelling capacity of soy proteins such as glycinin and β-conglycinin, noting that protein denaturation may represent a prerequisite for gel formation, typically achieved by applying heat treatment to soy prior to gelation [50]. In our study, sheep milk with added protein isolates was subjected to heat treatment before fermentation, which may also have influenced the structure and syneresis of the fermented milk. It is well established that in the case of SPI, gel formation is affected by factors such as pH, temperature, and ionic strength [51].

The heat-induced gelling properties of proteins play a critical role in food production. According to Klost et al. [52], both soluble and insoluble aggregates may be formed during heating. Pea proteins are typically classified by solubility into salt-soluble globulins, water-soluble albumins, ethanol-soluble prolamins, and alkali-soluble glutelins [43]. The type and distribution of aggregates strongly affect gel characteristics: a high dispersion of insoluble aggregates may weaken the gel network, resulting in less stable gels with reduced strength and water-holding capacity. Furthermore, during thermal treatment, pea legume proteins have been reported to form large, insoluble aggregates that hinder gel formation, whereas pea protein isolates with a higher proportion of vicilin are capable of producing firmer gels due to the formation of smaller intermediate aggregates upon heating [53,54,55].

Table 2 presents the color parameters of sheep milk fermented with *L. johnsonii*, both control and supplemented with different protein isolates. With respect to lightness (L*), the sample JS3, enriched with 3% SPI, exhibited the lowest L* value, indicating a darkening of the product. In contrast, the remaining samples (JW1.5, JW3, JS1.5, JB1.5, JB3) demonstrated significantly higher L* values compared to the CJ control.

The a* parameter (representing the green–red axis) was negative for most of the fermented milk samples, both control and supplemented, ranging from −0.09 in JB3 to −1.94 in CJ. Only the JS3 sample, with 3% SPI, showed a greater shift toward the red spectrum.

The b* parameter (representing the blue–yellow axis) was positive in all fermented milk samples. However, significantly lower b* values compared to the CJ control were found in JS1.5, JB1.5, and JB3, indicating a reduced yellow contribution.

A significant reduction in chroma (C*) was observed in JW1.5, JW3, JS1.5, JB1.5, and JB3, while hue angle (h°) values were significantly decreased in all sheep milk samples with added protein isolates compared to the CJ control.

The color parameters of milk fermented with *L. casei* are presented in Table 3. A significant decrease in lightness (L*) was observed only in the CS3 sample compared with the control. Most of the samples displayed negative a* values, indicating a contribution of green coloration. Similarly to the JS3 sample, the CS3 sample with 3% soy protein isolate exhibited a higher contribution of red tones. This can be explained by the presence of soybean pigments, specifically isoflavones. Soy contains three major types of isoflavones in four chemical forms: the aglycones daidzein, genistein, and glycitein; the β-glucosides daidzin, genistin, and glycitin; the acetyl-β-glucosides, including 6″-O-acetyl-β-daidzin, 6″-O-acetyl-β-genistin, and 6″-O-acetyl-β-glycitin; and the malonyl-β-glucosides, such as 6″-O-malonyl-β-daidzin, 6″-O-malonyl-β-genistin, and 6″-O-malonyl-β-glycitin [43]. The majority of isoflavones in soy and soy protein products—including defatted flour, isolates, concentrates, and textured proteins—occur in esterified forms, representing approximately 97–98% of the total [56].

Similar conclusions were reported by Pelaes Vital et al. [45], who also demonstrated that color was primarily influenced by the properties of soy, with soy-containing samples appearing darker, redder, and more yellow. Likewise, Gomes da Costa et al. [57] observed that yogurts enriched with protein were darker than the control yogurt without protein supplementation.

ANOVA analysis (Table 4) indicated that the color parameters of fermented milk were significantly affected by the type of protein isolate, the supplementation level, and the interactions between these two factors (*p* ≤ 0.01). In contrast, the three-way interaction between isolate type, bacterial strain, and isolate level did not significantly influence color.

### 3.2. Organoleptic Evaluation of Fermented Sheep Milk

The effect of supplementation with different protein isolates on the organoleptic properties of sheep milk fermented by *L. johnsonii* is illustrated in Figure 1. The consistency of fermented milk supplemented with WPI, particularly samples JW1.5 and JW3.0, received the highest scores from the panelists. These samples were characterized by a uniform and smooth texture without a gritty sensation. By contrast, the remaining fermented milk samples received slightly lower scores due to a noticeable graininess, especially in JS3 with 3% SPI. ANOVA analysis (Table 4) indicated that the consistency of fermented milk was significantly affected by both isolate type and concentration (*p* ≤ 0.01).

The addition of WPI at both 1.5% and 3% also enhanced the intensity of the milky-creamy flavor compared with the control and the other fermented samples. Neither the type of protein isolate nor the bacterial strain used for fermentation had a significant impact on sour taste perception. However, milk fermented with PPI (JB1.5 and JB3) was perceived as less sour than the CJ control. Supplementation with different isolates also intensified sour odor compared with the CJ control. It should be noted that panelists detected off-tastes in several samples, particularly JW3, JS1.5, JS3, and JB1.5, while the most pronounced off-taste and off-odor were recorded in JB3 with 3% PPI. This finding highlights the need to consider the use of flavoring agents in future formulations to mask the undesirable taste imparted by PPI.

Hashim et al. [22], in their study on fat-free yogurt enriched with whey protein isolate, also reported that WPI supplementation improved yogurt consistency. They emphasized that yogurts fortified with WPI achieved higher scores for structure, and that the addition of WPI (3% and 5%) markedly improved body and texture, as well as taste (3% WPI), compared to the control samples. These findings are in agreement with the observations made in our study regarding the organoleptic evaluation of WPI-fortified fermented milk. In contrast, Gomes da Costa et al. [57] reported that panelists found no differences in sensory attributes between non-enriched and protein-enriched yogurts, with the characteristics of each product being primarily associated with the presence of whey protein and the curdled texture.

The results of the organoleptic evaluation of sheep milk fermented with *L. casei* and supplemented with different protein isolates are presented in Figure 2. The addition of SPI and PPI negatively affected the texture of fermented milk, particularly in CS3 and CB3, compared with both the control and the other samples. According to the panelists, the milky-creamy taste of most protein-enriched samples was comparable to that of the control. The incorporation of protein isolates from different sources did not significantly influence sour taste, sour odor, or sweetness. However, in CS3 and CB3, the addition of 3% PPI produced the most pronounced off-taste and off-odor.

ANOVA analysis did not show a significant effect of the three studied factors (isolate type, isolate concentration, and bacterial strain) or their interactions on the perception of sour and sweet taste, as well as sour odor (Table 4).

Several statistically significant correlations were identified (Appendix A) between consistency and the color attributes (L*, a*, b*, C*, h°) of fermented milk. The milky–creamy taste was more intense in lighter samples (r = 0.32, *p* ≤ 0.05), in samples with lower syneresis (r = −0.35, *p* ≤ 0.05), and in those with better consistency (r = 0.43, *p* ≤ 0.05). By contrast, off-taste was positively associated with titratable acidity (r = 0.51, *p* ≤ 0.05) and occurred more frequently in samples with poorer consistency (r = −0.49, *p* ≤ 0.05). As expected, a significant correlation was also observed between off-taste and off-odor (r = 0.52, *p* ≤ 0.05), indicating that increases in off-taste intensity were accompanied by concomitant increases in off-odor intensity.

The sensory differences observed in our study likely arise from protein–water interactions and gel network architecture that depend on protein origin and dose. Whey proteins, after partial unfolding, probably form fine-stranded, disulfide-stabilized microgels with high water-holding and emulsifying capacity, which aligns with the higher consistency scores and reduced whey separation in WPI-fortified samples [58,59]. By contrast, legume globulins (soy/pea) tend to generate larger, less soluble heat-induced aggregates and more heterogeneous networks. At higher inclusion levels graininess and greater syneresis are more likely [60,61]. A moderate addition (1.5%) may optimize the balance between protein–protein and protein–water interactions, whereas excessive dosing can favor aggregation over hydration and thus weaken texture.

Flavor differences may also reflect matrix effects: plant-protein systems often carry characteristic legume volatiles (e.g., aldehydes), possibly modulating creamy/sour notes and contributing to off-notes at higher SPI/PPI [62]. Overall, compositional control of gel microstructure, via protein type and level, and the associated effects on pH and serum retention, appears to underpin the sensory outcomes [63].

Practical strategies can be integrated into protein-based carriers to temper beany/grassy notes. Recent literature details complementary approaches—ranging from enzymatic or heat-assisted treatments to LAB/yeast fermentation and β-cyclodextrin–based masking—applied across a variety of food products, including soy/pea systems and plant-based analogues. Targeted enzymatic hydrolysis, and, where appropriate, controlled glycation (protein–polysaccharide conjugation), can reduce the perception of key aldehydes (e.g., hexanal) via carbonyl–amine interactions and the release of antioxidant peptides; to limit the formation of such aldehydes in the first place, mild heat treatment to inactivate lipoxygenase (LOX) is recommended [64,65,66,67]. Microbial fermentation with selected lactic acid bacteria and food-grade yeasts has been shown to bioconvert aldehydes/ketones and generate masking esters and organic acids that rebalance aroma in pea/soy systems; process control (strain choice, pH, time/temperature) is essential to balance flavor benefits with probiotic viability [68,69]. In parallel, β-cyclodextrin inclusion complexes—where permitted—and compatible clean-label flavorants can bind grassy aldehydes and attenuate off-notes without compromising formulation simplicity [70,71]. As a practical complement, flavor additions (e.g., vanilla, cocoa, coffee, citrus oils, or fruit preparations) are widely used to attenuate grassy/beany notes without altering the protein matrix, and recent work shows that such aroma enrichment can improve palatability and consumer acceptance of soy/pea-based products [64,71].

### 3.3. Viability and Survival of Probiotic Bacteria

To exert beneficial health effects in the host, probiotic bacteria must survive passage through the gastrointestinal tract, demonstrating tolerance to low pH, bile salts, and gastric enzymes, followed by the ability to adhere to and colonize intestinal epithelial cells [72,73]. This functionality may be modulated by the type of food matrix in which probiotics are delivered [74]. Products with a pH above 5 and high buffering capacity can mitigate gastrointestinal acidity, thereby supporting microbial stability [75]. Moreover, food matrices can protect bacteria by limiting direct exposure to adverse gastrointestinal conditions, while some components may further interact with probiotics to influence their biological properties. In our study, we quantified the strength and direction of correlations (Appendix A) between pre-digestion probiotic cell counts and the physicochemical and sensory properties of the beverages. Probiotic counts were significantly correlated with pH and titratable acidity, consistent with prior reports in the literature.

To determine whether supplementation with protein isolates of different origins contributes to an increased number of viable probiotic cells (*L. johnsonii* or *L. casei*), microbiological analyses of fermented milk samples were performed after 7 days of refrigerated storage (Table 5 and Table 6). In addition, the number of viable *L. johnsonii* (Table 5) and *L. casei* cells (Table 6) was assessed at different stages of simulated digestion.

The viability data presented in Table 5 and Table 6 demonstrate a pronounced reduction in probiotic counts following the gastric phase, accompanied by a partial restoration of viability during the subsequent intestinal phase. Specifically, *L. casei* (Table 6) declined from an initial concentration exceeding 9 log CFU g^−1^ in the oral phase to as low as 2.40–3.43 log CFU g^−1^ after gastric digestion. However, bacterial counts increased during the intestinal phase, reaching levels between 4.05 and 4.71 log CFU g^−1^. A comparable trend was observed for *L. johnsonii* (Table 5), with post-gastric counts ranging from 4.37 to 7.00 log CFU g^−1^ and a subsequent increase to 4.00–7.30 log CFU g^−1^ following exposure to intestinal conditions. This trend, characterized by reduced viability under gastric conditions followed by improved recovery in the intestinal phase, has been widely reported in gastrointestinal simulation studies. The low pH of the gastric phase can severely impair probiotic cells, reducing their ability to grow on culture media, even though some remain metabolically active. When these injured cells enter the more favorable conditions of the intestinal phase (neutral pH, nutrient availability), many are able to restore membrane integrity and resume growth, which is reflected as an increase in colony-forming units. Govaert et al. [76] observed a similar effect during in vitro digestion and linked the regained culturability to milder pH and supportive matrix conditions. Likewise, Han et al. [77] emphasized that acid-damaged probiotics can recover during intestinal transit if conditions improve, particularly in the presence of a buffering food matrix.

The results demonstrated that protein isolates enhanced the viability of *L. johnsonii* compared with the control prior to digestion, with the highest counts observed in samples supplemented with PPI (JB1.5 and JB3). At the oral phase, viable cell numbers ranged from 9.17 log CFU g^−1^ (CJ) to 10.05 log CFU g^−1^ (JB1.5). At this stage, all protein-enriched samples maintained higher probiotic counts relative to the control. Since pH in the initial sections of the gastrointestinal tract plays a critical role in probiotic survival, and tolerance to digestive enzymes and acidic conditions is strain-dependent [78], further assessment was conducted during gastric digestion. Both at the gastric and intestinal phases, the highest viable cell counts were recorded in samples fermented with *L. johnsonii* and supplemented with PPI (JB1.5) and SPI (JS3) (Table 5).

Overduin et al. [44] reported that during gastric digestion, pea protein tended to precipitate temporarily, unlike the rapidly digested bovine whey protein, which remained in solution. The size of pea protein aggregates was smaller (50–500 µm) compared with those formed by bovine casein, the typical slow-digestible protein. Similarly, Mo [79] observed that the addition of 10% SPI to milk proteins enhanced digestibility at concentrations of 2.37 and 2.67 g/100 mL. This effect was likely related to limited aggregation between SPI and milk proteins during gastrointestinal digestion, as confirmed by microscopy. At a higher protein concentration (2.97 g/100 mL), however, this advantage was not apparent. Supplementation with 10% SPI slightly reduced the proportion of random coil structures at the end of gastric digestion, which in turn lowered the hydrolysis rate of whey proteins during the intestinal phase.

In another study, Pinho [80] showed that, under in vitro digestion, size-exclusion chromatography revealed that gastric conditions were insufficient to fully disrupt proteins and peptides embedded in gel networks. Although gastric digestion likely “opened” protein structures, mass transfer was limited by steric hindrance within the gel. At the intestinal stage, digestion proceeded more rapidly, probably due to the more porous nature of SPI-containing protein matrices compared with the denser particulate structure of WPI gels, which increased the accessibility of proteins to pancreatic enzymes.

As a commensal bacterium, *L. johnsonii* must withstand harsh conditions of low pH and high bile concentrations in the intestine to survive, colonize, multiply, and exert its beneficial functions [81]. To achieve this, the strain has developed multiple resistance and tolerance mechanisms, while also competing with other resident microbes in the same niche [82,83,84]. Its cells are surrounded by a compact proteinaceous S-layer, and additional protective structures include extracellular peptidoglycan, teichoic acids, and both capsular and exopolysaccharides, which contribute to cellular integrity and adhesion to the host. Stress-sensing mechanisms and export systems further complement its stress resistance machinery [85]. Moreover, *L. johnsonii* is highly adaptable to the host’s nutritional environment, as its genome encodes numerous phosphotransferase system (PTS) and ATP-binding cassette (ABC) transporters, along with amino acid proteases and peptidases that enable the utilization of a broad spectrum of sugars and amino acids available in the gastrointestinal tract [83,86].

Importantly, in our study, protein isolate supplementation positively influenced the viable counts of *L. johnsonii* at all stages of digestion. This may be explained in part by the strain’s biochemical activity, which includes the ability to metabolize substrates such as D-galactose, D-glucose, D-fructose, D-mannose, *N*-acetylglucosamine, arbutin, esculin, salicin, D-cellobiose, D-maltose, D-lactose, D-sucrose, D-raffinose, amygdalin, gentiobiose, and D-tagatose [59]. By comparison, *L. casei* has been shown to ferment galactose, glucose, lactose, fructose, mannose, mannitol, *N*-acetylglucosamine, and tagatose, but not glycerol, erythritol, arabinose, L-xylose, melibiose, raffinose, glycogen, xylitol, fucose, D-arabitol, or potassium 2- and 5-ketogluconate [87]. In addition, *L. casei* possesses the ability to utilize certain C5 sugar alcohols (e.g., adonitol), C5 sugars (e.g., ribose), and C6 sugar alcohols (e.g., sorbitol and dulcitol).

The application of different protein isolates also increased the viable counts of *L. casei* both prior to and at each stage of digestion (Table 6). This finding was confirmed by ANOVA analysis, which demonstrated a significant effect (*p* ≤ 0.001) of isolate type on probiotic cell numbers throughout the gastrointestinal tract (Table 4).

Before digestion and at the oral phase, *L. casei* counts exceeded 9 log CFU g^−1^ across all fermented milk samples, with the highest values recorded in CS1.5 and CS3. During gastric digestion, viable counts of *L. casei* ranged from 2.40 log CFU g^−1^ in the CC control to 3.66 log CFU g^−1^ in CS1.5. After passage through the stomach, probiotics must survive the conditions of the small intestine, where they are exposed to pancreatin, bile salts, and a pH of approximately 8.0. Similarly to gastric tolerance, intestinal survival of probiotic bacteria is also strongly influenced by the carrier food matrix. In our study, supplementation with protein isolates of different origins enhanced the viability of *L. casei* during the intestinal phase compared with the control. The highest increases were observed in samples with WPI (CW1.5, CW3) and SPI (CS1.5, CS3).

ANOVA analysis (Table 4) further indicated that intestinal cell counts were significantly influenced by all studied factors (type of isolate, bacterial strain, and isolate concentration) as well as their interactions (*p* ≤ 0.01).

Maintaining high probiotic viability and activity in fermented foods and during gastrointestinal transit is essential for ensuring their health-promoting efficacy. Protein isolates incorporated into fermented sheep milk may exert multiple functions: they provide nutrients for probiotic metabolism, create protective matrices, buffer acidic environments to mitigate digestive stresses, and shield probiotic cells from environmental challenges [30,88].

Protein matrices likely support probiotic robustness through nutritional and structural contributions. When exogenous protein constitutes the main nitrogen source, many lactic acid bacteria depend on cell-envelope proteolysis and peptide uptake to secure essential amino acids, which may reinforce growth and stress responses under acidic and biliary challenges [89,90]. Beyond nutrient supply, the surrounding protein network can moderate exposure to harsh luminal conditions by slowing acid/bile ingress and potentially buffering the immediate microenvironment of the cells. This interpretation is consistent with the broader microencapsulation literature, in which biopolymer walls, commonly proteins or polysaccharides such as whey protein isolate (WPI), soy protein isolate, alginate, pectin or gum arabic, frequently improve survival during simulated gastrointestinal transit and enhance storage stability [91,92,93,94]. In comparative settings, WPI-based systems have often shown especially durable protection under refrigerated storage and in acid/bile challenges [95]. Effective probiotic action further requires adhesion and persistence in the gut. An adequate nutrient supply can facilitate proliferation, whereas firm adhesion contributes to mucosal integrity, epithelial signaling, and competitive exclusion of undesirable microbes [96,97]. Finally, survival appears strain-specific: different taxa vary in acid/bile tolerance and in bile-salt hydrolase activity, which could explain divergent outcomes under otherwise similar matrix conditions [98]. Altogether, these lines of evidence provide a plausible link between protein origin/structure, matrix microarchitecture, and strain-dependent survival patterns observed in our study, while situating the results within established frameworks of protein-mediated encapsulation and protection.

Recent work indicates that protein-based microcapsules can meaningfully enhance probiotic delivery. Emulsion encapsulation of *Lactobacillus reuteri* with whey protein concentrate and gum arabic achieved high encapsulation efficiency and greater stability than free cells during storage and simulated digestion, likely due to robust interfacial films and partial pH buffering by the protein shell [99]. Reviews similarly highlight dairy and plant proteins (whey, casein, hempseed, pea) as versatile wall materials that allow tunable permeability and timed intestinal release [32]. Moreover, pea-protein systems—e.g., double W/O/W emulsions reinforced with cellulose nanocrystals—improved survival of *Lacticaseibacillus rhamnosus* GG under simulated digestion, with pea-protein microencapsulation maintaining higher viability than non-encapsulated cells in recent in vitro models [100,101]. Together, these advances suggest that engineered dairy- and plant-protein matrices can be combined with prebiotic polysaccharides to strengthen gels and enhance probiotic delivery in fermented milk.

Figure 3 and Figure 4 illustrate the survival rates of *L. johnsonii* and *L. casei*, expressed as percentages. The addition of protein isolates had a significant positive effect on the survival of *L. johnsonii*. The highest survival (>70%) was recorded in samples supplemented with WPI (JW1.5 and JW3) and SPI (JS3), while the lowest survival (43.10%) was observed in the CJ control (Figure 3). For *L. casei*, survival was most influenced by the type of protein isolate used. The highest values, approximately 50%, were obtained in samples CW1.5 and CW3 with WPI, whereas in the remaining treatments survival exceeded 40% (Figure 4).

Consistent with these findings, Cordeiro et al. [102] reported that supplementation of skim milk with 30% WPI improved the survival of *L. casei* BL23 under highly acidic conditions (pH 2 for 60 min) and in the presence of bile salts, compared with the unsupplemented control. WPI increased the survival of *L. casei* from 55–69% (in unsupplemented milk) to as high as 80% in supplemented milk, thereby enhancing tolerance to both acid and bile salts. This supports our observation that WPI may provide effective protection under simulated digestion, not only due to its physical properties (gelling capacity, resistance to pepsin) but also through the creation of a nutrient-rich, buffered microenvironment [103].

All protein isolates tested in this study serve as sources of amino acids and peptides that can be utilized by lactic acid bacteria [30,104]. The incorporation of WPI, SPI, or PPI into the fermentation medium not only promotes microbial growth and metabolic activity during fermentation but also improves stress resistance during simulated gastrointestinal passage. The availability of readily assimilable proteins and peptides enhances probiotic nutrition, thereby increasing cell resilience under stress conditions such as gastric acidity and bile exposure [104]. Consequently, protein isolates may act as “protein-based prebiotics,” supporting both probiotic survival during in vitro digestion and their potential colonization in the intestine. However, the efficiency of this mechanism depends on strain-specific characteristics. In particular, *L. johnsonii* LJ exhibited a superior ability to utilize isolate-derived substrates compared with *L. casei* 431, which likely contributed to its enhanced cell protection and higher survival rates following simulated digestion.

All protein isolates tested in this study provide lactic acid bacteria with amino acids and peptides that can be readily utilized [105]. Including WPI, SPI, or PPI in the fermentation medium not only supports bacterial growth and metabolic activity during fermentation but also enhances probiotic stress tolerance during simulated gastrointestinal transit. The resulting abundance of readily assimilable proteins and peptides improves probiotic nutrition, thereby increasing cellular resilience under harsh conditions such as exposure to gastric acidity and bile [105]. Consequently, protein isolates may function as “protein-based prebiotics,” supporting probiotic survival during in vitro digestion and potentially aiding their colonization in the intestine. However, the efficacy of this protective mechanism appears to be strain-dependent. Notably, *L. johnsonii* LJ exhibited a greater capacity to utilize isolate-derived substrates than *L. casei* 431, which likely contributed to its superior cell protection and higher survival rates following simulated digestion.

Beyond nutrient supply, the intrinsic structure of the proteins used for fortification governs microgel formation and, in turn, the protective function of the delivery matrix during digestion. Whey proteins are globular (e.g., β-lactoglobulin, α-lactalbumin) and, upon heat/acid treatment, undergo partial unfolding and disulfide-mediated aggregation into heat-set microgels with tunable particle size and network porosity. Such networks increase water-holding and create diffusion barriers that slow acid and bile ingress to embedded cells. Recent reviews and in situ digestion studies link these whey-derived microstructures with improved stabilization of sensitive bioactives and probiotics during gastric transit [106,107]. In contrast, caseins are largely intrinsically disordered and organized as micelles; acidification promotes micellar rearrangement and para-casein network formation with pH-dependent coagulation and buffering, yielding gels that can physically entrap cells and moderate local acidity during gastric coagulation [108,109]. For plant proteins, the dominant storage globulins determine gel morphology. In soy, glycinin (11S) tends to form stronger, more elastic gels, whereas β-conglycinin (7S) produces more viscous, weaker networks; preheating and the 11S/7S ratio modulate unfolding, hydrophobic interactions and disulfide exchange, thereby controlling gel strength and pore size relevant to probiotic protection [110,111]. In pea protein, the legumin/vicilin balance and ionic environment (e.g., Ca^2+^) steer heat-induced gelation and microstructure–rheology coupling; gels with finer pores and higher water-holding capacity are associated with improved mechanical integrity and reduced diffusional flux, features considered favorable for maintaining culturability under gastric–intestinal stress [112].

Collectively, these structure–function relationships rationalize the strain-dependent survival patterns observed here: matrices forming cohesive, fine-pored protein microgels (e.g., whey- or legumin-rich systems, or acid-gelled casein networks) are more likely to buffer pH, limit proton/bile diffusion, and preserve membrane integrity, thereby supporting higher recovery of injured cells during the intestinal phase [106,108].

The choice between plant and whey protein isolates entails trade-offs. Recent life-cycle assessments generally report lower GHG emissions, land use, and water use for plant protein ingredients, although extraction and fractionation steps may attenuate these advantages depending on process energy and coproduct allocation [113,114,115]. By contrast, whey proteins typically provide higher protein quality (DIAAS ≥ 100 in adults), whereas many plant isolates exhibit lower DIAAS unless complemented via blending or targeted amino-acid fortification [116,117]. For functional dairy formulations, these observations suggest a pragmatic pathway: leverage plant isolates where environmental gains are meaningful, maintain or restore protein quality through formulation (e.g., whey–plant blends or leucine enrichment), and verify that such choices preserve probiotic performance.

## 4. Conclusions

Sensory and technological outcomes of fermented sheep milk were composition-dependent and can be plausibly attributed to matrix effects on gel microstructure, protein–water interactions, and serum retention. Whey and plant isolates produced distinct network architectures that shaped consistency, syneresis, flavor profile, and the tolerance of *Lacticaseibacillus casei* and *Lactobacillus johnsonii* during simulated gastrointestinal passage. Moderate supplementation generally favored a balance between protein–protein and protein–water interactions, whereas higher inclusion was more likely to promote aggregation and textural trade-offs. These results support a formulation-led approach in which the type and level of protein isolate are selected to target specific textural and probiotic objectives while managing sensory constraints associated with plant proteins. Further work should refine plant-protein utilization and extend strain coverage to confirm matrix–strain specificity under broader application conditions.

## Figures and Tables

**Figure 1 nutrients-17-03340-f001:**
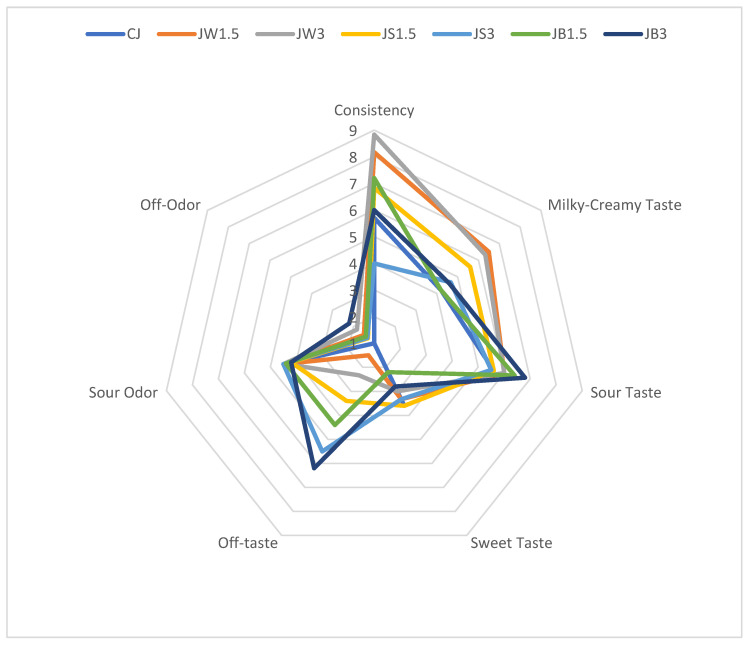
Organoleptic parameters of milk fermented by *L. johnsonii* with protein isolates. CJ—control sample; JW1.5/JW3—1.5%/3% WPI; JS1.5/JS3—1.5%/3% SPI; JB1.5/JB3—1.5%/3% PPI.

**Figure 2 nutrients-17-03340-f002:**
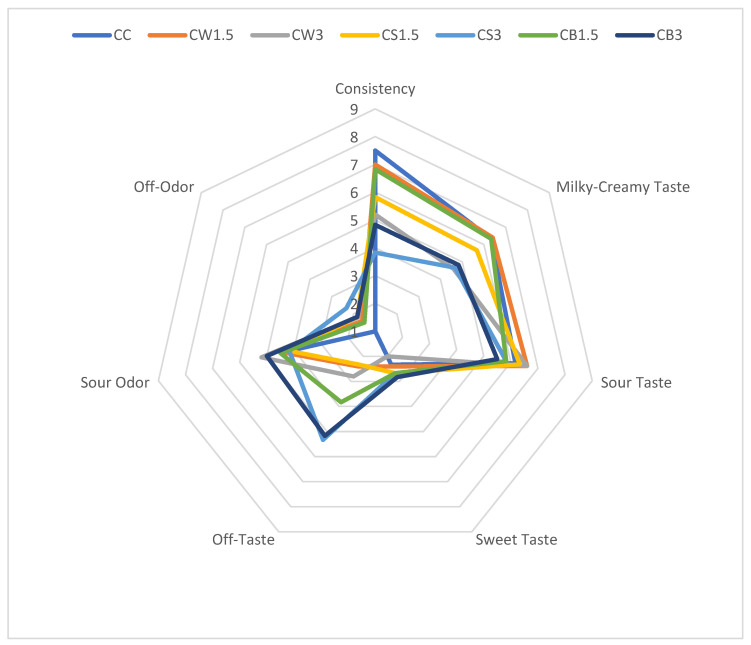
Organoleptic parameters of milk fermented by *L. casei* with protein isolates. CC—control sample; CW1.5/CW3—1.5%/3% WPI; CS1.5/CS3—1.5%/3% SPI; CB1.5/CB3—1.5%/3% PPI.

**Figure 3 nutrients-17-03340-f003:**
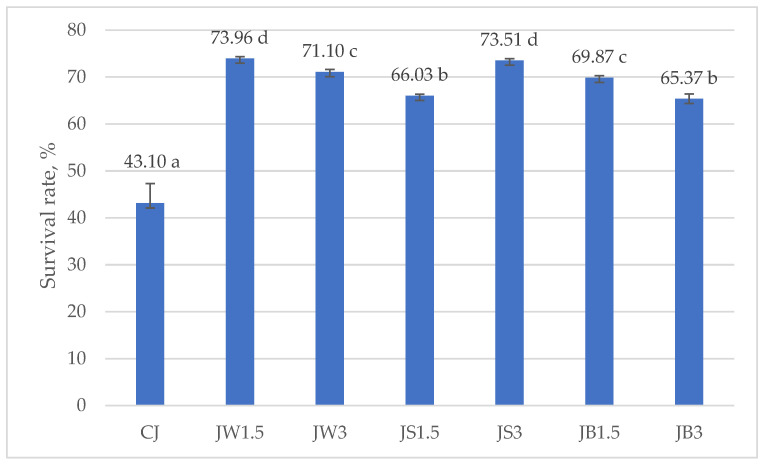
Survival rates in milk fermented by *L. johnsonii*. a–d—mean values denoted by different letters differ statistically significantly at *p* ≤ 0.05; CJ—control sample; JW1.5/JW3—1.5%/3% WPI; JS1.5/JS3—1.5%/3% SPI; JB1.5/JB3—1.5%/3% PPI.

**Figure 4 nutrients-17-03340-f004:**
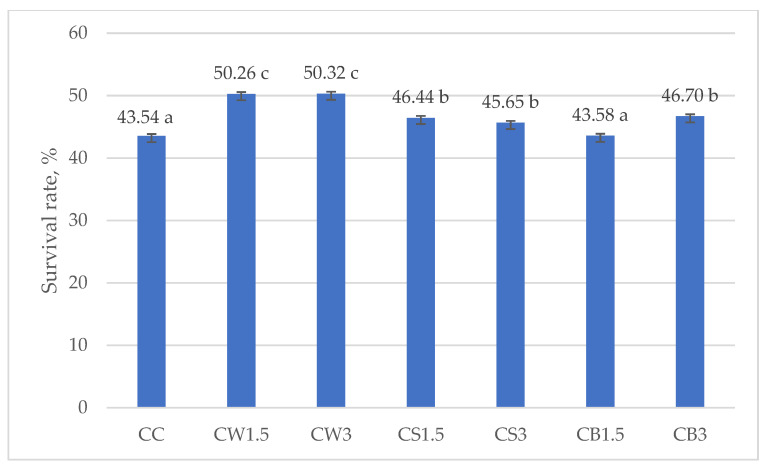
Survival rates in milk fermented by *L. casei*. a–c—mean values denoted by different letters differ statistically significantly at *p* ≤ 0.05; CC—control sample; CW1.5/CW3—1.5%/3% WPI; CS1.5/CS3—1.5%/3% SPI; CB1.5/CB3—1.5%/3% PPI.

**Table 1 nutrients-17-03340-t001:** Experimental groups.

Probiotic Strain	Experimental Groups						
	Control	1.5% WPI	3% WPI	1.5% SPI	3% SPI	1.5% PPI	3% PPI
*L. casei* 431	CC	CW1.5	CW3	CS1.5	CS3	CB1.5	CB3
*L. johnsonii* LJ Delvo^®^Pro	CJ	JW1.5	JW3	JS1.5	JS3	JB1.5	JB3

WPI—whey protein isolate; SPI—soy protein isolate; PPI—pea protein isolate.

**Table 2 nutrients-17-03340-t002:** Properties of sheep milk with protein isolates fermented by *Lactobacillus johnsonii*.

Properties	CJ	JW1.5	JW3	JS1.5	JS3	JB1.5	JB3
pH 1	6.67 ^d^ ±0.01	6.52 ^b^ ±0.02	6.45 ^a^ ±0.03	6.68 ^d^ ±0.02	6.70 ^d^ ±0.03	6.64 ^d^ ±0.02	6.60 ^c^ ±0.01
pH 7	4.29 ^e^ ±0.03	4.20 ^c^ ±0.01	4.24 ^d^ ±0.01	4.22 ^d^ ±0.01	4.17 ^b^ ±0.01	4.14 ^a^ ±0.01	4.13 ^a^ ±0.01
L	91.18 ^b^ ±1.02	94.51 ^c^ ±1.13	94.77 ^c^ ±0.67	94.67 ^c^ ±1.91	87.97 ^a^ ±1.90	95.59 ^c^ ±2.59	93.26 ^c^ ±1.03
a*	−1.91 ^a^ ±0.20	−1.34 ^b^ ±0.12	−1.25 ^b^ ±0.05	−1.24 ^b^ ±0.25	2.01 ^d^ ±0.21	−0.52 ^c^ ±0.46	−0.09 ^c^ ±0.09
b*	11.48 ^b^ ±0.32	10.62 ^ab^ ±1.26	11.01 ^b^ ±0.14	10.40 ^a^ ±0.32	12.59 ^b^ ±1.64	10.47 ^a^ ±0.26	10.22 ^a^ ±0.52
C*	11.63 ^b^ ±0.34	10.01 ^a^ ±0.27	11.08 ^a^ ±0.14	10.48 ^a^ ±0.29	12.75 ^b^ ±1.62	10.49 ^a^ ±0.29	9.33 ^a^ ±0.39
h°	99.76 ^e^ ±0.66	95.69 ^c^ ±4.04	96.59 ^d^ ±0.41	96.82 ^cd^ ±1.52	80.84 ^a^ ±1.34	93.11 ^c^ ±2.64	90.51 ^b^ ±0.52
TA %	1.18 ^a^ ±0.06	1.26 ^b^ ±0.02	1.25 ^b^ ±0.05	1.35 ^c^ ±0.01	1.51 ^d^ ±0.01	1.56 ^de^ ±0.05	1.60 ^e^ ±0.02
Syneresis, %	41.28 ^e^ ±1.35	5.45 ^a^ ±1.92	17.52 ^d^ ±1.26	5.97 ^a^ ±1.60	8.01 ^b^ ±1.48	9.59 ^b^ ±1.98	13.19 ^c^ ±1.79

^a–e^ Means within a row followed by different letters differ significantly at *p* ≤ 0.05; CJ—control sample; JW1.5/JW3—1.5%/3% WPI; JS1.5/JS3—1.5%/3% SPI; JB1.5/JB3—1.5%/3% PPI.

**Table 3 nutrients-17-03340-t003:** Properties of sheep milk with protein isolates fermented by *Lacticaseibacillus casei*.

Properties	CC	CW1.5	CW3	CS1.5	CS3	CB1.5	CB3
pH 1	6.67 ^d^ ±0.01	6.54 ^b^ ±0.02	6.48 ^a^ ±0.03	6.66 ^d^ ±0.02	6.70 ^d^ ±0.03	6.62 ^c^ ±0.02	6.61 ^c^ ±0.01
pH 7	4.34 ^a^ ±0.01	4.37 ^b^ ±0.01	4.39 ^b^ ±0.01	4.41 ^c^ ±0.02	4.44 ^d^ ±0.01	4.35 ^a^ ±0.01	4.41 ^c^ ±0.01
L*	95.91 ^b^ ±1.03	95.82 ^b^ ±0.70	95.15 ^b^ ±1.42	95.77 ^b^ ±2.04	91.16 ^a^ ±1.61	95.88 ^b^ ±0.34	94.81 ^ab^ ±1.12
a*	−1.80 ^a^ ±0.17	−1.66 ^ab^ ±0.21	−1.42 ^b^ ±0.11	−1.07 ^c^ ±0.17	1.93 ^e^ ±0.86	−0.67 ^d^ ±0.54	−0.40 ^d^ ±0.11
b*	10.34 ^ab^ ±0.60	11.02 ^b^ ±0.41	10.90 ^b^ ±0.46	10.97 ^b^ ±0.59	10.93 ^ab^ ±1.28	10.03 ^a^ ±0.45	10.68 ^ab^ ±1.11
C*	10.49 ^a^ ±0.62	11.06 ^b^ ±0.41	11.00 ^b^ ±0.47	11.03 ^b^ ±0.59	11.78 ^b^ ±2.36	10.08 ^a^ ±0.44	9.69 ^a^ ±1.11
h°	99.87 ^d^ ±0.53	98.57 ^cd^ ±0.94	97.42 ^c^ ±0.65	96.14 ^bc^ ±0.88	91.59 ^a^ ±1.83	95.71 ^b^ ±0.40	92.39 ^a^ ±0.73
TA %	1.10 ^ab^ ±0.05	1.12 ^b^ ±0.01	1.08 ^ab^ ±0.07	1.04 ^a^ ±0.03	1.06 ^a^ ±0.04	1.11 ^ab^ ±0.01	1.03 ^a^ ±0.01
Syneresis, %	23.16 ^e^ ±0.73	16.73 ^c^ ±1.46	21.32 ^d^ ±1.48	14.16 ^b^ ±0.93	20.00 ^d^ ±0.66	11.98 ^a^ ±1.27	18.55 ^c^ ±1.48

^a–e^ Means within a row followed by different letters differ significantly at *p* ≤ 0.05; CC—control sample; CW1.5/CW3—1.5%/3% WPI; CS1.5/CS3—1.5%/3% SPI; CB1.5/CB3—1.5%/3% PPI.

**Table 4 nutrients-17-03340-t004:** ANOVA analysis—effects of the type and dose of protein isolate, bacterial strain, and their interactions on the properties of fermented sheep milk.

Properties		Type of Isolate	Bacterial Strain	Dose of Isolate	Type of Isolate × Bacterial Strain	Type of Isolate × Dose of Isolate	Bacterial Strain × Dose of Isolate	Type of Isolate × Dose of Isolate × Bacterial Strain
pH 1		xx	ns	xx	ns	xx	ns	ns
pH 7		x	xx	ns	ns	ns	xx	xx
L		xx	x	xx	ns	xx	xx	ns
a*		xx	ns	xx	ns	xx	xx	ns
b*		xx	ns	xx	ns	xx	xx	ns
C*		xx	ns	x	ns	xx	ns	ns
h°		xx	ns	xx	ns	xx	ns	ns
TA %		xx	xx	xx	xx	xx	x	xx
Syneresis %		ns	x	xx	xx	ns	xx	x
Consistency		xx	ns	xx	ns	ns	x	ns
Milky-Creamy Taste		ns	ns	xx	ns	ns	ns	ns
Sour Taste		ns	ns	ns	ns	ns	ns	ns
Sweet Taste		ns	ns	ns	ns	ns	ns	ns
Off-Taste		xx	ns	xx	ns	ns	ns	ns
Sour Odor		ns	ns	ns	ns	ns	ns	ns
Off-Odor		ns	ns	x	ns	ns	ns	ns
Stages of Digestion	Before Digestion	xx	x	ns	xx	x	x	x
	Oral Cavity	xx	xx	ns	xx	xx	ns	xx
	Stomach	xx	xx	ns	xx	xx	ns	xx
	Small Intestine	xx	xx	xx	xx	xx	xx	xx

x—significant at *p* ≤ 0.05; xx—significant at *p* ≤ 0.01; ns—not significant.

**Table 5 nutrients-17-03340-t005:** Cell count of *Lactobacillus johnsonii* (log CFU g^−1^) in fermented sheep milk depending on the stage of digestion.

Stages of Digestion	CJ	JW1.5	JW3	JS1.5	JS3	JB1.5	JB3
Before Digestion	9.28 ^aA^ ±0.19	9.87 ^cA^ ±0.21	9.69 ^bcA^ ±0.23	9.45 ^bA^ ±0.10	9.78 ^bcA^ ±0.55	10.09 ^dA^ ±0.16	10.08 ^dA^ ±0.17
Oral Cavity	9.17 ^aA^ ±0.21	9.64 ^cA^ ±0.15	9.68 ^cA^ ±0.19	9.33 ^bA^ ±0.11	9.65 ^cA^ ±0.19	10.05 ^dA^ ±0.20	10.04 ^dA^ ±0.22
Stomach	3.84 ^aC^ ±0.13	6.79 ^cC^ ±0.10	6.86 ^cB^ ±0.16	4.37 ^bC^ ±0.28	6.80 ^cC^ ±0.01	7.00 ^dB^ ±0.18	4.66 ^bC^ ±0.30
Small Intestine	4.00 ^aB^ ±0.01	7.30 ^dB^ ±0.32	6.89 ^cB^ ±0.08	6.24 ^bB^ ±0.12	7.19 ^dB^ ±0.27	7.05 ^dB^ ±0.15	6.59 ^bB^ ±0.29

^a–d^ Means within a row followed by different letters differ significantly at *p* ≤ 0.05; ^A–C^ Means within a column followed by different letters differ significantly at *p* ≤ 0.05; CJ—control sample; JW1.5/JW3—1.5%/3% WPI; JS1.5/JS3—1.5%/3% SPI; JB1.5/JB3—1.5%/3% PPI.

**Table 6 nutrients-17-03340-t006:** Cell count of *Lacticaseibacillus casei* (log CFU g^−1^) in fermented sheep milk depending on the stage of digestion.

Stages of Digestion	CC	CW1.5	CW3	CS1.5	CS3	CB1.5	CB3
Before Digestion	9.30 ^aA^	9.31 ^aA^	9.32 ^aA^	10.14 ^cA^	10.23 ^cA^	9.98 ^cA^	9.55 ^bA^
	±0.20	±0.13	±0.28	±0.29	±0.19	±0.23	±0.15
Oral Cavity	9.19 ^aA^	9.21 ^aA^	9.20 ^aA^	10.11 ^cA^	10.10 ^cA^	9.91 ^cA^	9.45 ^bA^
	±0.44	±0.09	±0.36	±0.23	±0.22	±0.28	±0.15
Stomach	2.40 ^aC^	3.20 ^cC^	3.43 ^cC^	3.66 ^cC^	2.97 ^bC^	2.86 ^bC^	2.88 ^bC^
	±0.12	±0.19	±0.16	±0.21	±0.10	±0.12	±0.11
Small Intestine	4.05 ^aB^	4.68 ^cB^	4.69 ^cB^	4.71 ^cB^	4.67 ^cB^	4.35 ^bB^	4.46 ^bB^
	±0.16	±0.13	±0.09	±0.18	±0.14	±0.18	±0.13

^a–c^ Means within a row followed by different letters differ significantly at *p* ≤ 0.05; ^A–C^ Means within a column followed by different letters differ significantly at *p* ≤ 0.05; CC—control sample; CW1.5/CW3—1.5%/3% WPI; CS1.5/CS3—1.5%/3% SPI; CB1.5/CB3—1.5%/3% PPI.

## Data Availability

The original contributions presented in this study are included in the article. The raw data supporting the conclusions—including unprocessed measurement results—are available from the authors upon reasonable request (mpawlos@ur.edu.pl, aznamirowska@ur.edu.pl).

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
