# Peer review of "Probiotic Sheep Milk: Physicochemical Properties of Fermented Milk and Viability of Bacteria Under Simulated Gastrointestinal Conditions"

_nutrients, 2025, doi:10.3390/nu17213340_

Round 1
Reviewer 1 Report
Comments and Suggestions for Authors
Thank you for the opportunity to participate in the review of the manuscript titled "Probiotic Sheep Milk: Physicochemical Properties of Fermented Milk and Viability of Bacteria under Simulated Gastrointestinal Conditions."
The manuscript describes studies of fermented sheep milks, their technological properties, and their survival during simulated gastrointestinal conditions.
The manuscript follows the typical structure of a research article. It consists of well-defined chapters. The introduction to the topic is well-written and provides essential information about sheep milk, probiotic bacteria, and protein fortification. The literature cited in the introduction is new and relevant. The Materials and Methods section is correct. The proposed studies are relevant to the topic. The division of the study samples is clear. I request that the authors address the organoleptic assessment and the lack of information regarding the ethics committee's approval for human studies. The presentation of the results raises no objections, and the discussion is appropriate to the research topic.
In summary, the manuscript is well-written. It is suitable for Nutrients after minor corrections and clarification of a few issues.
Detailed comments:
Line 9. Abstract. What is the purpose of the study? Please clarify and add it to the abstract.
Line 17, 111. Please add strain symbols to the bacterial species.
Line 33. Please change the keywords so that they do not overlap with the words in the manuscript title. This will increase the chances of finding the article in databases.
Line 118. Please add the purpose of the study at the end of introduction.
Line 177. Please add the subject. “The separated … was”
Line 193. How many color measurements were performed? Please clarify.
Line 194. Do the authors have the approval of the ethics committee for organoleptic tests performed by humans? Did the procedure comply with the Declaration of Helsinki?
Line 243. Please clarify whether the fermented milk is given in g or mL. Previous subsections use mL, now use g.
Line 245. Please clarify whether this refers to the number of probiotic bacteria? The number of mesophilic lactic acid bacteria is determined on MRS agar. Does sheep's milk, as a raw material, contain LAB? The milk was heat-treated. Was the LAB count checked after heat treatment, and were these bacteria undetectable? In my opinion, the LAB count can be the sum of the bacteria present in the milk plus the added bacteria, not just the probiotic bacteria. Please clarify.
Line 270. I suggest organizing the "Results and Discussion" section by introducing subsections (similar to the "Methods"). This will make the manuscript easier to read. Furthermore, tables and graphs with results should ideally be placed directly below the paragraph where they are cited.
Figures 1 and 2. Organoleptic evaluation graphs are not legible. Please enlarge the graph and add more numbers to the scale so that the differences between samples can be seen.
Tables 5 and 6. Please explain the phenomenon of low LAB counts after gastric passage, followed by higher LAB counts after intestinal passage.
Author Response
Response to Reviewer 1 Comments
Thank you very much for taking the time to review this manuscript. Please find the detailed responses below and the corresponding corrections highlighted in the re-submitted files. For your convenience, all changes have been highlighted in yellow throughout the revised manuscript.
Line 9. Abstract. What is the purpose of the study? Please clarify and add it to the abstract.
We have slightly revised the sentence to state the study aim more explicitly:
L15-17: The aim of this study was to clarify the functional potential of such formulations by assessing probiotic survival under in vitro digestion simulating oral, gastric, and intestinal phases.
Line 17, 111. Please add strain symbols to the bacterial species.
We have added the specific strain symbols for each probiotic. The two strains used are Lacticaseibacillus casei 431 and Lactobacillus johnsonii LJ (L18, L149, L150).
Line 33. Please change the keywords so that they do not overlap with the words in the manuscript title. This will increase the chances of finding the article in databases.
We have revised the keyword list to remove redundancy with the title and to improve indexing. “Sheep milk” was removed, as it already appears in the title. The updated list now includes terms emphasizing the main experimental variables and analytical aspects of the study:
probiotic bacteria; in vitro digestion; whey protein isolate; soy protein isolate; pea protein isolate; functional food; physicochemical properties; microbiological analysis (L35-36).
Line 118. Please add the purpose of the study at the end of introduction.
We have added an explicit statement of the study aim at the end of the Introduction. The revised text now reads:
L145: The aim of this study was to evaluate the effect of protein isolates fortification on the physicochemical and sensory properties of fermented sheep milk and on the survival of probiotic bacteria during simulated gastrointestinal digestion.
Line 177. Please add the subject. “The separated … was”
We have corrected the sentence by adding the missing subject. It now reads:
L215-217: The separated whey was collected and weighed, and syneresis was calculated as the percentage of the original sample mass represented by the supernatant.”
Line 193. How many color measurements were performed? Please clarify.
The explanation regarding the number of measurements is provided in Section 2.9 (Statistical Analysis). As stated there, each experimental condition was conducted in triplicate, with five technical replicates per trial. This applies also to the color analysis described in Section 2.5.
Line 194. Do the authors have the approval of the ethics committee for organoleptic tests performed by humans? Did the procedure comply with the Declaration of Helsinki?
Please find the English translations of the informed consent form and the ethical approval statement attached. The originals were in Polish, reflecting that all participants on the sensory panel were Polish.
The organoleptic tests were carried out using standard sensory evaluation methods, in which participants assessed safe food products prepared exclusively from ingredients approved for human consumption. The study involved no invasive procedures and did not pose any health risk to participants. According to national regulations and institutional policy, such sensory tests do not require approval from the Ethics Committee or Institutional Review Board.
The study did not involve any biomedical intervention or collection of personal data; therefore, the Declaration of Helsinki is not applicable to this type of sensory evaluation.
Line 243. Please clarify whether the fermented milk is given in g or mL. Previous subsections use mL, now use g.
We confirm that the amount of fermented milk used in the in vitro digestion assay (L267) was measured by weight (g). The notation has been clarified in the revised manuscript to ensure consistency between sections.
Line 245. Please clarify whether this refers to the number of probiotic bacteria? The number of mesophilic lactic acid bacteria is determined on MRS agar. Does sheep's milk, as a raw material, contain LAB? The milk was heat-treated. Was the LAB count checked after heat treatment, and were these bacteria undetectable? In my opinion, the LAB count can be the sum of the bacteria present in the milk plus the added bacteria, not just the probiotic bacteria. Please clarify.
The Methods already specify that microbiological enumeration quantified the viable cells of the inoculated probiotic strains (Lacticaseibacillus casei 431 and Lactobacillus johnsonii LJ). As described, all formulations (including those containing protein isolates) prepared with commercial organic sheep milk (“I Love My Sheep,” Leeb Biomilch GmbH, Austria) were heat-treated at 85 °C for 10 min and cooled prior to inoculation. Counts were obtained by pour-plating on MRS agar under anaerobic conditions; therefore, the reported values reflect only the added probiotic cultures.
Line 270. I suggest organizing the "Results and Discussion" section by introducing subsections (similar to the "Methods"). This will make the manuscript easier to read. Furthermore, tables and graphs with results should ideally be placed directly below the paragraph where they are cited.
We appreciate the reviewer’s valuable suggestion. The Results and Discussion section has been reorganized into three clearly defined subsections to improve readability and alignment with the structure of the Materials and Methods section. The revised subsections are as follows:
3.1. Physicochemical Properties of Fermented Sheep Milk
3.2. Organoleptic Evaluation of Fermented Sheep Milk
3.3. Viability and Survival of Probiotic Bacteria
Additionally, all tables and figures have been repositioned to appear directly below the corresponding paragraphs where they are first discussed.
Figures 1 and 2. Organoleptic evaluation graphs are not legible. Please enlarge the graph and add more numbers to the scale so that the differences between samples can be seen.
We have implemented the requested improvements to Figures 1 and 2 (L491, L550).
Tables 5 and 6. Please explain the phenomenon of low LAB counts after gastric passage, followed by higher LAB counts after intestinal passage.
Thank you for the comment. A detailed explanation of this phenomenon has been added to the Discussion section (Section 3.3), with reference to the viability data presented in Tables 5 and 6 (L574-L592).

Reviewer 2 Report
Comments and Suggestions for Authors
The manuscript presents a well-designed and clearly written study that makes a valuable contribution to the field of functional dairy science. The topic is timely and relevant, and the results are supported by sound experimental methods and robust statistical analysis. The work successfully demonstrates the potential of sheep milk fortified with various protein isolates as an effective carrier for probiotics under simulated gastrointestinal conditions.
To further strengthen the paper and enhance its scientific depth, several refinements are recommended:
Add p-values or significance levels directly in figure captions or tables to facilitate easier interpretation of the statistical outcomes.
Expand the discussion on how protein structure (globular vs. fibrous; plant vs. dairy origin) affects microgel formation and the protective role of the matrix in maintaining probiotic viability during digestion.
Integrate more recent literature (post-2022) concerning protein-based matrices and probiotic delivery systems to place the findings in the context of the most up-to-date research.
Discuss potential strategies to mitigate the beany off-flavor associated with soy and pea proteins-such as enzymatic hydrolysis, fermentation with flavor-masking yeasts, or the use of natural flavor additives.
Include an explicit sensory acceptance threshold, for example, indicating the minimum mean score required for consumer marketability.
Provide more methodological precision, specifying the exact enzyme activities used in the digestion model and justifying the chosen incubation times.
Clarify how temperature and pH were controlled and monitored throughout the in vitro digestion procedure to ensure reproducibility.
Add a comparative perspective by including references or data on bovine and goat milk systems to better highlight the unique buffering and compositional properties of sheep milk.
Discuss briefly the sustainability and nutritional implications of using plant-based versus whey protein isolates in functional dairy formulations, which would further increase the paper’s relevance to current food innovation trends.
Author Response
Response to Reviewer 2 Comments
Thank you very much for taking the time to review this manuscript. Please find the detailed responses below and the corresponding corrections highlighted in the re-submitted files. For your convenience, all changes have been highlighted in green throughout the revised manuscript and accompanying materials.
To further strengthen the paper and enhance its scientific depth, several refinements are recommended:
Add p-values or significance levels directly in figure captions or tables to facilitate easier interpretation of the statistical outcomes.
Thank you for your comment. Information on statistical significance (e.g., “a–c Means within a row followed by different letters differ significantly at p ≤ 0.05”) was already included in the captions of all tables and figures, except for the organoleptic evaluation graphs.
Expand the discussion on how protein structure (globular vs. fibrous; plant vs. dairy origin) affects microgel formation and the protective role of the matrix in maintaining probiotic viability during digestion.
We have expanded the Discussion to address how protein structure (globular whey vs. micellar casein; dairy vs. plant) drives microgel formation and matrix‐mediated protection during digestion. The new paragraph (L764-789) explains whey microgels, acid-gelled casein networks, and plant-protein gelation rules (soy 11S/7S; pea legumin/vicilin, Ca²⁺) and links these mechanisms to our strain-dependent outcomes. This section is supported by recent literature (Jiang 2024; Bayrak 2023; Liao 2024; Runthala 2023; Huang 2025; Zhang 2024; Ren 2024).
Integrate more recent literature (post-2022) concerning protein-based matrices and probiotic delivery systems to place the findings in the context of the most up-to-date research.
We have integrated post-2022 literature on protein-based matrices and probiotic delivery systems. Representative additions include: Żulewska 2025; Gantumur 2024; Afzal 2024; Vanare 2025; Saiz-Gonzalo 2025; Teymoori 2024; Jiang 2024; Bayrak 2023; Liao 2024; Runthala 2023; Huang 2025; Zhang 2024; Ren 2024.
Discuss potential strategies to mitigate the beany off-flavor associated with soy and pea proteins-such as enzymatic hydrolysis, fermentation with flavor-masking yeasts, or the use of natural flavor additives.
We have revised the Discussion to include a concise paragraph outlining practical strategies to mitigate beany/grassy notes—covering enzymatic or heat-assisted treatments, LAB/yeast fermentation, β-cyclodextrin masking, and flavor additions—supported by recent literature (e.g., Yang, 2023; Sun et al., 2024; Nam et al., 2024; Tao et al., 2022; Zipori et al., 2024; Flores et al., 2024; Lee et al., 2020; Kelanne et al., 2024; Jaeger et al., 2024) (L481-500).
Include an explicit sensory acceptance threshold, for example, indicating the minimum mean score required for consumer marketability.
We have updated the Organoleptic Evaluation (Materials and Methods) to include an explicit sensory acceptance threshold. Specifically, we now state that a mean panel score ≥ 6.0 on the 9-point hedonic scale is considered the minimum level for consumer marketability of a sample.
L240-242: For the 9-point hedonic scale, overall sensory acceptability was interpreted as marketable when the mean overall liking was ≥ 6.0. Scores ≥ 7.0 were interpreted as indicative of high acceptability [36,37].
Provide more methodological precision, specifying the exact enzyme activities used in the digestion model and justifying the chosen incubation times.
The in vitro digestion model applied in this study followed the standardized static INFOGEST protocol (Minekus et al., 2014; Brodkorb et al., 2019), with minor adjustments for fermented milk matrices. The process simulated three physiological phases: oral, gastric, and intestinal.
During the oral phase, 50 mL of each fermented milk sample was mixed with 5 mL of simulated salivary fluid containing 2.38 g Na₂HPO₄, 0.19 g K₂HPO₄, 8.0 g NaCl, 100 mg mucin, and α-amylase with an activity of 200 U/L. The pH was adjusted to 6.75 ± 0.20, and the samples were incubated for 2 min at 37 °C with constant stirring (90 rpm).
In the gastric phase, pepsin (13.08 mg per vessel, corresponding to approximately 41,000–58,000 U total activity based on the supplier’s specification) was added, and the pH was adjusted to 2.00 ± 0.20 with 12 M HCl. The mixtures were incubated for 2 h at 37 °C with continuous stirring (90 rpm).
The intestinal phase was initiated by adding 5 mL of pancreatin solution (4 g/L) and 5 mL of bile salt solution (25 g/L). The pH was adjusted to 7.00 ± 0.20 using 1 M NaOH, and the digestion continued for 2 h at 37 °C (90 rpm). The pancreatin concentration provided a proteolytic activity equivalent to approximately 100 U/mL trypsin, comparable to the standard INFOGEST recommendations.
The incubation times—2 min (oral), 2 h (gastric), and 2 h (intestinal)—were selected according to the INFOGEST consensus, reflecting physiologically relevant residence times of semi-solid foods in the human gastrointestinal tract. These parameters ensured enzymatic activity levels and digestion kinetics comparable to those achieved under standardized conditions.
Minekus, M.; Alminger, M.; Alvito, P.; Ballance, S.; Bohn, T.; Bourlieu, C.; Carrière, F.; Boutrou, R.; Corredig, M.; Dupont, D.; et al. (2014). A Standardised Static In Vitro Digestion Method Suitable for Food—An International Consensus. Food & Function, 5, 1113–1124.
Brodkorb, A., Egger, L., Alminger, M. et al. INFOGEST static in vitro simulation of gastrointestinal food digestion. Nat Protoc 14, 991–1014 (2019).
Clarify how temperature and pH were controlled and monitored throughout the in vitro digestion procedure to ensure reproducibility.
We have clarified in the Materials and Methods section how temperature and pH were controlled during the digestion procedure.
L260-266: All digestion steps were performed under strictly controlled temperature and pH conditions to ensure reproducibility. The temperature was continuously maintained at 37 ± 0.5 °C using a thermostatically controlled shaking water bath (Orbital Shaker-Incubator ES 20, Biosan, Riga, Latvia). The pH of each digestion vessel was measured and recorded at the beginning and end of every phase using a calibrated pH electrode (InLab® Solids Pro-ISM electrode, Mettler-Toledo, Greifensee, Switzerland) connected to a FiveEasy™ pH meter (Mettler-Toledo, Greifensee, Switzerland).
Add a comparative perspective by including references or data on bovine and goat milk systems to better highlight the unique buffering and compositional properties of sheep milk.
To provide a clearer comparative context, we added a brief paragraph in the Introduction that contrasts bovine and caprine systems with ovine milk (L54-65). The paragraph is supported by recent sources (e.g., Arrichiello, 2022; Li et al., 2023; Ahlborn et al., 2023; Dinkçi et al., 2023).
Discuss briefly the sustainability and nutritional implications of using plant-based versus whey protein isolates in functional dairy formulations, which would further increase the paper’s relevance to current food innovation trends.
To address sustainability and nutritional implications, we added a brief paragraph to the end of the Discussion (L791-801).
Reviewer 3 Report
Comments and Suggestions for Authors
It is a well-written manuscript that matches the scope of Nutrition. To improve the presentation of the manuscript, it is suggested to re-write the abstract and avoid the use of titles (Methods, Results, Conclusions) and follow the instructions for authors.
Please describe in detail the fermented product; is it liquid or solid? Does any layer of fat appear on the surface? It is confusing since the title is for fermented milk and in the Discussion it is compared with yogurt results.
The main weakness is the statistical analysis of the sensory results; this needs to be described in detail. The use of ANOVA assumes a normal distribution and similar variance among samples; was it the case in the current study? Was the sensory evaluation repeated?
Specific comments:
L92, please explain what is meant by probiotic survival; is it in the food or in the GI system?
L122, was salt added in the milk?
L147, please clarify the medium, was it MRS or milk-based medium?
L177, separated or supernatant?
L273, heat treatment instead of pasteurization
Author Response
Response to Reviewer 3 Comments
Thank you very much for taking the time to review this manuscript. Please find the detailed responses below and the corresponding corrections highlighted in the re-submitted files. For your convenience, all changes have been highlighted in dark green throughout the revised manuscript.
It is a well-written manuscript that matches the scope of Nutrition. To improve the presentation of the manuscript, it is suggested to re-write the abstract and avoid the use of titles (Methods, Results, Conclusions) and follow the instructions for authors.
According to the official Instructions for Authors of Nutrients, original research articles are expected to include a structured abstract with the following headings: Background/Objectives, Methods, Results, and Conclusions. Therefore, we have retained the current structure to comply with journal guidelines.
Please describe in detail the fermented product; is it liquid or solid? Does any layer of fat appear on the surface? It is confusing since the title is for fermented milk and in the Discussion it is compared with yogurt results.
The fermented product examined in our study was a semi-liquid matrix obtained by fermenting commercially available homogenized sheep milk. In addition, the milk–protein isolate mixtures were also subjected to homogenization prior to fermentation. As a result of this dual homogenization process, no visible fat layer formed on the surface of the final fermented milk samples.
The term “fermented milk” is used in this manuscript in accordance with the Codex Alimentarius Standard for Fermented Milks (CODEX STAN 243-2003). We intentionally do not refer to the product as “yogurt,” since, by definition, yogurt refers to a fermented milk produced using a symbiotic culture of Streptococcus thermophilus and Lactobacillus delbrueckii subsp. bulgaricus.
The main weakness is the statistical analysis of the sensory results; this needs to be described in detail. The use of ANOVA assumes a normal distribution and similar variance among samples; was it the case in the current study?
We verified the ANOVA assumptions (normality and homogeneity of variance). They were satisfied, so ANOVA was used.
Was the sensory evaluation repeated?
The sensory evaluation was conducted with a trained panel of 30 individuals (15 female, 15 male) who assessed the samples using a standardized 9-point hedonic scale. Importantly, the entire experiment was repeated three times, i.e., n = 3 independent biological replicates per formulation, and in each replicate, all 30 assessors evaluated the samples under identical conditions. This ensures reproducibility and statistical robustness of the panel data.
We have revised the description of methods in the Statistical Analysis section:
L312-325: Results are reported as mean ± standard deviation (SD). Statistical analyses were performed in Statistica v13.1 (StatSoft, Tulsa, OK, USA). Physicochemical measurements were obtained in triplicate (n = 3 biological replicates per formulation), each measured in five technical replicates per trial; technical replicates were averaged within each biological replicate prior to analysis. Sensory data were collected from 30 assessors (n = 30; 15 female, 15 male) in each of three independent experimental trials (biological replicates), using a 9-point hedonic scale. Results are presented as mean ± SD of the individual assessor ratings for each formulation. Microbiological counts were generated for n = 3 biological replicates per formulation at each analysis, with duplicate plates per dilution; duplicate plates were averaged within each biological replicate. One-way and factorial ANOVA were used to evaluate treatment effects across all measured parameters, including individual sensory attributes. Tukey’s HSD was applied for multiple mean comparisons (p ≤ 0.05). Pearson correlation coefficients (r) were calculated between selected physicochemical, sensory, and microbiological variables, with statistical significance set at p ≤ 0.05.
Specific comments:
L92, please explain what is meant by probiotic survival; is it in the food or in the GI system?
The “probiotic survival” is used in two contexts in this paper: viability in the product during refrigerated storage, and survival during simulated gastrointestinal transit. We have corrected the manuscript:
L106-108: Protein fortification provides additional opportunities to enhance both product quality and probiotic survival, i.e. viability during refrigerated storage and survival under simulated gastrointestinal conditions.
L122, was salt added in the milk?
The sheep milk used was a retail product (“I Love My Sheep,” Leeb Biomilch GmbH, Austria), and the compositional values listed in L122 (now L160) (including 0.16% salt) were taken directly from the product label. In accordance with EU nutrition-labelling practice, the “salt” value on labels represents a salt equivalent calculated from sodium (i.e., salt = sodium × 2.5), which reflects naturally occurring sodium/chloride in the milk.
L147, please clarify the medium, was it MRS or milk-based medium?
Thank you for pointing this out. Pre-activation was performed in sheep milk, not in MRS. MRS was used only for plate enumeration. We have corrected the manuscript:
L185-186: Each culture was pre-activated in sheep milk at 40 °C for 5 h [34] to achieve a high viable cell count at inoculation.
L177, separated or supernatant?
Thank you for catching this wording error. It was a typographical mistake. We have corrected the sentence:
L215-217: The separated whey was collected and weighed, and syneresis was calculated as the percentage of the original sample mass represented by the supernatant.”
L273, heat treatment instead of pasteurization
We have corrected the wording at L203 and L329: the term “pasteurization” has been replaced with “heat treatment.”
Reviewer 4 Report
Comments and Suggestions for Authors
This study investigates the effect of protein isolate fortification (whey, soy, and pea) on the physicochemical, organoleptic, and probiotic survival properties of fermented sheep milk inoculated with Lacticaseibacillus casei and Lactobacillus johnsonii. The work combines food technology, microbiology, and nutritional functionality, highlighting the interaction between matrix composition and probiotic viability under simulated gastrointestinal conditions. Here are my observations:
The Introduction is comprehensive, logically structured, and well-referenced. The research gap is mentioned but could be more focused (“lack of studies integrating plant and animal protein isolates in probiotic sheep milk”). Emphasize the novelty.
The Materials and Methods section is detailed and reproducible; all reagents, enzyme sources, and incubation parameters clearly stated. Colorimetric and sensory analyses described with high precision, but statistical treatment of panel data not indicated (no ANOVA per sensory attribute). Include reproducibility (%RSD) or coefficient of variation for microbiological counts.
Physicochemical and Organoleptic Properties section systematically presented with full statistical treatment (Tables 2–4). Discussion sometimes descriptive rather than analytical (repeats data without mechanistic insight). Figures 1–2 (sensory results) could be integrated into a single comparative panel. Link sensory outcomes to compositional effects. Include a correlation analysis between pH, syneresis, and sensory consistency.
Microbiological Viability and In Vitro Digestion section summarize cell counts and survival rates. Figures 3–4 lack standard deviation/error bars; only letters for significance are shown. There is Limited mechanistic interpretation how protein structure or amino acid composition affects survival is only briefly touched. Discussion does not relate to prior probiotic encapsulation or protein–matrix protection literature. Integrate a short paragraph comparing strain-dependent survival trends.
The Conclusions are overly general and slightly repetitive of abstract.
Author Response
Response to Reviewer 4 Comments
Thank you very much for taking the time to review this manuscript. Please find the detailed responses below and the corresponding corrections highlighted in the re-submitted files. For your convenience, all changes have been highlighted in light blue throughout the revised manuscript.
The Introduction is comprehensive, logically structured, and well-referenced. The research gap is mentioned but could be more focused (“lack of studies integrating plant and animal protein isolates in probiotic sheep milk”). Emphasize the novelty.
We sharpened the research gap to emphasize novelty. In the Introduction, we now state that, to our knowledge, no prior study has integrated both animal (whey) and plant (soy/pea) protein isolates within a single fermented sheep-milk model while jointly assessing physicochemical quality and probiotic survival across a standardized in vitro digestion (L140-145).
The Materials and Methods section is detailed and reproducible; all reagents, enzyme sources, and incubation parameters clearly stated. Colorimetric and sensory analyses described with high precision, but statistical treatment of panel data not indicated (no ANOVA per sensory attribute). Include reproducibility (%RSD) or coefficient of variation for microbiological counts.
Thank you for this helpful suggestion. We have now computed Pearson correlation coefficients between selected physicochemical, sensory, and microbiological variables and added a new Supplementary Table (Table S1) with the correlation matrix (L565-568).
We have revised the description of methods in the Statistical Analysis section:
L312-325: Results are reported as mean ± standard deviation (SD). Statistical analyses were performed in Statistica v13.1 (StatSoft, Tulsa, OK, USA). Physicochemical measurements were obtained in triplicate (n = 3 biological replicates per formulation), each measured in five technical replicates per trial; technical replicates were averaged within each biological replicate prior to analysis. Sensory data were collected from 30 assessors (n = 30; 15 female, 15 male) in each of three independent experimental trials (biological replicates), using a 9-point hedonic scale. Results are presented as mean ± SD of the individual assessor ratings for each formulation. Microbiological counts were generated for n = 3 biological replicates per formulation at each analysis, with duplicate plates per dilution; duplicate plates were averaged within each biological replicate. One-way and factorial ANOVA were used to evaluate treatment effects across all measured parameters, including individual sensory attributes. Tukey’s HSD was applied for multiple mean comparisons (p ≤ 0.05). Pearson correlation coefficients (r) were calculated between selected physicochemical, sensory, and microbiological variables, with statistical significance set at p ≤ 0.05.
Physicochemical and Organoleptic Properties section systematically presented with full statistical treatment (Tables 2–4). Discussion sometimes descriptive rather than analytical (repeats data without mechanistic insight). Figures 1–2 (sensory results) could be integrated into a single comparative panel. Link sensory outcomes to compositional effects. Include a correlation analysis between pH, syneresis, and sensory consistency.
We sincerely thank the reviewer for this helpful suggestion regarding Figures 1 and 2. While we agree that combining both sensory datasets into a single comparative panel could facilitate side-by-side interpretation, we respectfully prefer to retain the current two-figure format. This is because the number of sample variants included in each analysis is relatively large, and overlaying multiple data series on a single plot would likely reduce visual clarity and readability. Presenting the datasets separately ensures that individual trends and differences remain easy to interpret. We hope the reviewer will find this approach acceptable.
We performed the requested correlation analysis between pH, syneresis, and sensory consistency. The results are provided in Table S1 and are described and interpreted in the Discussion (L506-514).
We have expanded the Discussion with an analytical paragraph that links the sensory outcomes to compositional variables and physicochemical factors, and provides mechanistic context (protein–water interactions, gel network architecture, heat-induced aggregation, and legume-derived volatiles).
L515-524: The sensory differences observed in our study likely arise from protein–water interactions and gel network architecture that depend on protein origin and dose. Whey proteins, after partial unfolding, probably form fine-stranded, disulfide-stabilized microgels with high water-holding and emulsifying capacity, which aligns with the higher consistency scores and reduced whey separation in WPI-fortified samples [Alting, 2000; Vasbinder et al., 2004]. By contrast, legume globulins (soy/pea) tend to generate larger, less soluble heat-induced aggregates and more heterogeneous networks. At higher inclusion levels graininess and greater syneresis are more likely [Sun and Arntfield, 2012; Loveday et al., 2020]. A moderate addition (1.5%) may optimize the balance between protein–protein and protein–water interactions, whereas excessive dosing can favor aggregation over hydration and thus weaken texture.
Flavor differences may also reflect matrix effects: plant-protein systems often carry characteristic legume volatiles (e.g., aldehydes), possibly modulating creamy/sour notes and contributing to off-notes at higher SPI/PPI (Sim et al., 2021). Overall, compositional control of gel microstructure, via protein type and level, and the associated effects on pH and serum retention, appears to underpin the sensory outcomes (Rodríguez et al., 2023).
Microbiological Viability and In Vitro Digestion section summarize cell counts and survival rates. Figures 3–4 lack standard deviation/error bars; only letters for significance are shown.
Thank you for pointing this out. Figures 3–4 have been updated to include error bars (±SD).
There is Limited mechanistic interpretation how protein structure or amino acid composition affects survival is only briefly touched. Discussion does not relate to prior probiotic encapsulation or protein–matrix protection literature. Integrate a short paragraph comparing strain-dependent survival trends.
We have expanded the Discussion to provide a clearer mechanistic interpretation of how protein structure and amino-acid provision may influence probiotic survival, explicitly linking our findings to prior encapsulation and protein–matrix protection literature (L680-713).
The Conclusions are overly general and slightly repetitive of abstract.
We have revised the Conclusions to remove general statements and avoid overlap with the Abstract (L803-814).
Round 2
Reviewer 3 Report
Comments and Suggestions for Authors
Accept in present form
Reviewer 4 Report
Comments and Suggestions for Authors
The article has been improved it's ready for publication.